# Unprecedented quiescence in resource development area allows detection of long-lived latent seismicity

Rebecca O. Salvage[1] and David W. Eaton[1]

[1]Dept. of Geoscience, University of Calgary, Calgary, AB, T2N 1N4

**Correspondence:** Rebecca O. Salvage (beckysalvage@gmail.com)

**Abstract.**

Recent seismicity in Alberta and north-east British Columbia has been attributed to ongoing oil and gas development in the area, due to its temporal and spatial correlation. Prior to such development, the area was seismically quiescent. Here, we show evidence that latent seismicity may occur in areas where previous operations have occurred, even during a shutdown in operations. The global pandemic of COVID-19 furnished the unique opportunity to study seismicity during a long period of anthropogenic quiescence. 389 events were detected within the Kiskatinaw area of British Columbia from April to August 2020, which encompasses a period with very little hydraulic fracturing operations. This reduction in operations was the result of a government imposed lockdown severely restricting the movement of people, as well as a downturn in the economic market causing industry stock prices to collapse. Except for a reduction in the seismicity rate and a lack of temporal clustering that is often characteristic of hydraulic fracturing induced sequences, the general characteristics of the observed seismicity were similar to the preceding time period of active operations. During the period of relative quiescence, event magnitudes were observed between $M_L$ -1 and $M_L$ 1.2, which is consistent with previous event magnitudes in the area. Hypocenters occurred in a corridor orientated NW-SE, just as seismicity had done in previous years, and located at depths associated with the target Montney formation or shallower ($<$2.5 km). A maximum of 21% of the detected events during lockdown may be attributable to natural seismicity, with a further 8% potentially attributed to dynamic triggering of seismicity from teleseismic events and 6% related to ongoing salt-water disposal and a single operational well pad. However, this leaves $\sim$65% of the seismicity detected during lockdown being unattributable to primary activation mechanisms. This seismicity is unlikely to be the result of direct pore pressure increases (as very little direct injection of fluids was occurring at the time) and we see no patterns of temporal or spatial migration in the seismicity as would be expected from direct pore pressure increases. Instead, we suggest that this latent seismicity may be generated by aseismic slip as fluids (resulting from previous hydraulic fracturing injection) become trapped within permeable formations at depth, keeping pore pressures in the area elevated, and consequently allowing the generation of seismicity. Alternatively, this seismicity may be the result of fault and fracture weakening in response to previous fluid injection. This is the first time that this latent seismicity has been observed in this area of British Columbia, and as such, this may now represent the new normal background seismicity rate within the Kiskatinaw area.

## 1 Introduction

The number of recorded instances of injection-induced seismicity has risen dramatically over the past decade, in part due to increased operations in hydraulic fracturing, waste-water disposal and enhanced geothermal systems around the globe, as well as enhanced monitoring meaning we are better able to detect smaller events (e.g. Atkinson et al., 2016; Ellsworth, 2013). In western Canada, the Western Canadian Sedimentary Basin (WCSB) is the focus of such activity, where a number of distinct resource plays are located including the Montney and the Duvernay. Despite an apparent flurry of larger magnitude seismic events associated with these operations (e.g. $M_L$ 4.5 near Fort St John, British Columbia in November 2018, Babaie-Mahani et al. (2019); Peña Castro et al. (2020); $M_W$ 4.1 near Fox Creek, Alberta in January 2016, Eyre et al. (2019b)) very few hydraulic fracturing operations (∼0.8%) are actually linked to seismic activity with $M_W > 3$ (Ghofrani and Atkinson, 2020).

The Montney Play, which is Lower-Middle Triassic in age, is formed of extensive fine-grained siliclastic units (inter-bedded sand, silt and mudstones), and stretches from west-central Alberta to north-east British Columbia (Eaton and Schultz, 2018; Dixon, 2000; Armitage, 1962). Over 5,600 multistage horizontal hydraulically fractured wells had been completed within the Montney by December 2018 (Nieto et al., 2018). In recent years, north-east British Columbia has experienced an increasing number of felt seismic events during active development within the Montney play. This led the British Columbia Oil and Gas Commission to implement a special order in 2018, within the area now known as the Kiskatinaw Seismic Monitoring and Mitigation Area (KSMMA), which required operators to undertake a pre-assessment of the seismic hazard, fully inform residents in the area of upcoming operations and undertake real-time seismic monitoring before, during and after completions (BC Oil and Gas Commission, 2018). Of particular importance was the introduction of the threshold for the suspension of operations following events with ≥M3 within the KSMMA, which is lower than the ≥M4 threshold that is standard elsewhere in British Columbia (e.g. Babaie-Mahani and Kao, 2020).

Prior to the introduction of oil and gas extraction, western Canada was generally seismically quiet, except for the Mackenzie Mountains and the North American plate boundary off the west coast of British Columbia (Lamontagne et al., 2008). Consequently seismicity detected within the KSMMA has been assumed to be directly related to ongoing oil and gas operations due to its temporal and spatial correlation with active wells. However, there are a number of examples of seismicity thought to be related to hydraulic fracturing that generate events months after operations have ceased (e.g. Eyre et al., 2020). We call this latent seismicity i.e. seismicity that appears after an unusually long delay following a primary activation processes, but which has no obvious "trigger" (e.g. enhanced pressurization at the onset of seismicity), and which cannot be explained by other sources (e.g. natural or dynamic triggering processes).

Here, we investigate seismicity generated within the KSMMA during the unprecedented period of quiescence that resulted from the global COVID-19 pandemic. The slowing of operations in the area and the reduction in seismic noise as businesses shut down and people stayed indoors, compounded by a downturn in the energy market, gave us the unique opportunity to

study latent seismicity in an area where it would usually go undetected. This study is highly unusual considering that this was a near-complete shutdown of operations in the area, rather than one dictated by a short suspension in operations within a specific area (under the traffic light system in KSMMA following an event of ≥M3) or the cessation of seismicity following reservoir depletion. Given that prior to the development of the Montney play this area was relatively quiet in terms of natural seismicity (Lamontagne et al., 2008), the detection of latent seismicity over ∼4 months suggests lingering changes in the stress field to allow for its generation.

## 2  COVID-19 and the reduction of noise globally

The year 2020 was highly unusual due to the global pandemic that caused the shutdown of many businesses and severely restricted the movement of people worldwide. This reduction in ground motion has been accurately measured by a drop in seismic ambient noise in many places, and correlated with a decrease in population mobility (e.g. Lecocq et al., 2020; Dias et al., 2020). The noise level at a seismic station can be estimated using the probabilistic power spectral density (PPSD) of its records (McNamara and Buland, 2004). Following the methodology of Lecocq et al. (2020), we compute the PPSD from 30-minute windows with 50% overlap so that a single value is gained for each window, calculated using Welch's method (Welch, 1967) for the vertical component at different seismic stations. We use the vertical component since a number of public stations (including station R25AC, Fig. 1), are single component seismometers, enabling us to compare the reduction in noise across a number of sensors. This method reduces numerical noise in the power spectra at the expense of reducing the frequency resolution because of frequency binning, but this effect is minimized with a robust smoothing parameterization. The 30-minute time series are then converted to an average daily PSD, and the RMS of the time-domain displacement is extracted. Anthropogenic cultural noise typically concentrates at high frequencies (>1-10 Hz, McNamara and Buland (2004)), but is strongly diurnal (e.g. stronger during the day than at night, and stronger during the weekdays compared to the weekends (Lecocq et al., 2020)). To capture this noise, but avoid meteorological signals, and in particular to avoid focusing on oceanic microseisms (which typically manifest below 1 Hz), we use the frequency band of 4-14 Hz to investigate seismic noise during the pandemic.

Figure 1 shows the reduction of seismic noise in the frequency band 4-14 Hz in Gastown, Vancouver, British Columbia during the global pandemic. A clear reduction in noise is observed following the closure of schools (black line) and businesses (red line). During Phase I of the pandemic i.e. between the closure of businesses and the partial reopening of the city on 5 May 2020 (green line; red background colour), noise levels remain lower than previously recorded. Following the re-opening of some businesses in May and June 2020, the increase in noise is interpreted as the increased movement of people, although it remains lower than pre-pandemic levels. To verify that these variations do not occur on an annual basis, we undertook the same noise analysis for the year 2019, and found no such fluctuations during the corresponding months. In fact, the average ground displacement remained between 20 and 30 nm at station R25AC for the entirety of 2019.

## 3 Seismicity in the KSMMA

With increasing unconventional oil and gas operations within the KSMMA over the past decade, the number of public monitoring stations has also increased. Prior to 2020, 9 public sensors maintained by Natural Resources Canada, the British Columbia Oil and Gas Commission, the British Columbia Seismic Research Consortium, and the Geological Survey of Canada existed within the KSMMA boundary, along with 6 co-located accelerometers poised to better capture higher levels of ground motion from larger seismic events. In early 2020, 13 additional broadband seismic stations (Trillium T120 seismometers with Taurus digitizers, sampling rate of 200 Hz) and two Titan accelerometers (sampling rate of 200 Hz) were installed within the KSMMA (expanding the EON-ROSE (EO) network) as part of a joint project between the University of Calgary, Nanometrics, Geoscience BC and a number of universities in South Korea, to monitor ongoing seismicity associated with hydraulic fracturing operations (Fig. 2). Sensors were installed at existing well sites through the support of four companies, and placed just below the surface (to a depth of 30 cm) to enhance coupling and decrease noise. The primary aim of the installation in 2020 was to expand monitoring capabilities within the KSMMA, in particular in the north-east and south-west of the area where prior public monitoring was sparse. The sites of the accelerometers were chosen due to their proximity to the most recent seismicity in the area, in particular a number of felt events that have occurred close to Tower Lake and Farmington. Further details of the installation of this array can be found in Salvage et al. (2021).

The catalogue of seismic events detected in the KSMMA is based on the newly installed array and available public stations in the area. Events were detected from the incoming continuous seismic data using an STA/LTA triggering algorithm, followed by a template-matching algorithm. The template-matching algorithm uses previously detected events from its own catalogue, as well as historical seismicity in the area to identify seismic events from seismic noise, and consequently remove unwanted signals. A machine learning technique was then used to identify phase arrivals within suspected events, using historical seismicity (from both public and private arrays) as a training database. By converting the waveforms into over 250 "features" (e.g. frequency content, P-S timings), machine learning enabled the association of such features with P and S phases (or conversely, with noise), allowing accurate automatic phase picking of detected events (Salvage et al., 2021). We take the catalogue of event times and P and S phases to cut waveforms from the continuous seismic data. Data are band-passed filtered between 1 Hz and 80 Hz (zero-phase), converted to displacement by removing the instrument response, and de-trended using both the mean of the waveform and fitting a linear function to the waveform with a least-squares. We then determine hypocenter locations using NonLinLoc (Lomax et al., 2009, 2000), a probabilistic, global-search non-linear algorithm that generates the maximum likelihood hypocenter location based on the estimated posterior probability density function for each event. A 1D velocity model, specifically calibrated for the KSMMA from compressional and shear sonic logs, formation tops and ground truth locations of previous seismicity (available directly from the British Columbia Oil and Gas Commission), was used for location analysis. Events were then re-located using HypoDD, a double difference algorithm, whereby the residual between the observed and calculated travel-time difference (or double-difference) between two earthquakes observed on a single station are related to differences in their relative hypocenter locations and origin times (Waldhauser and Ellsworth,

2000). To calculate magnitudes we use a form of the Richter (1935) magnitude formula that has been modified to better reflect local attenuation characteristics within the KSMMA (Babaie-Mahani and Kao, 2020). In line with calculations conducted by Natural Resources Canada (NRCan), we calculated $M_L$ using the maximum amplitude from the vertical component simulated on a Wood-Anderson (WA) seismometer, rather than the horizontal component, which has been used elsewhere.

Historically, seismicity within the KSMMA appears to occur within spatially distinct regions that fall within a corridor orientated NW-SE. Figure 2 shows the detected seismicity within the KSMMA in 2018 (Fig. 2(a), Visser et al. (2020)), compared to 2020 (Fig. 2(b), Salvage et al. (2021)). We are unable to compare the seismicity from 2019 since these data are not yet published. In both years, the largest magnitude event occurred in an area away from the densest occurrence of seismicity. Since the largest event in 2020 did not occur in the same cluster as the largest event of 2018, it appears that the occurrence

of $M_L$ 3+ events, i.e. those that may be felt, is not necessarily confined to a single region. Temporally, seismicity within the KSMMA occurs in distinct clusters, attributed to ongoing development activity in the area (Fig. 3). In 2018, heightened periods of seismicity were observed in April, May, July and August (Fig. 3(a)). Similar periods of heightened seismicity were observed in 2020 in March, August and September (Fig. 3(b)). The majority of seismicity detected within the KSMMA is $M_L \leq 2$, and consequently goes unfelt.

## 3.1   Prior and Post Lockdown: 2020

In March 2020, the Province of British Columbia introduced measures aimed at slowing the spread of COVID-19, including the closure of schools and childcare facilities on 17 March, and the closure of many businesses (in particular those that included daily human interaction) on 21 March. Although these closures did not include a government enforced closure of hydraulic fracturing activities, the stalling economy led to stock prices of operators within the KSMMA plummeting to record lows,

and as such the suspension of most activities (S. Venables, British Columbia Oil and Gas Commission, Pers. Comm., January 2021). Up until this point in 2020, similar patterns of seismicity to other years were observed in the KSMMA (Fig. 3). A total of 4,268 events were detected from the onset of data collection (22 January 2020) from the updated EO array (yellow triangles, Fig. 2) to 1 April. Following the initial closure of businesses on 21 March, there is evidence of ongoing hydraulic fracture operations for ∼10 days, with associated heightened seismicity (Fig. 3(b)). This reflects the continuation of planned

operations by companies within the KSMMA, following which, no new operations were initiated due to the diminishing economy. Magnitudes of recorded seismicity prior to lockdown at the end of March range from $M_L$ -0.7 to $M_L$ 2.9.

At the beginning of April, a period of relative seismic quiescence began in the KSMMA (Fig. 3(b); grey background). The COVID-19 pandemic not only limited the movement of people and shut businesses, but also caused an economic downturn

in the energy sector, leading to the cancellation of many operations. Seismicity since the resumption of activities in the later summer months, ∼4 months after the lockdown began, is once again temporally clustered, with a total of 3,176 events being recorded from 6 August 2020 to 1 January 2021. The largest magnitude event of 2020 occurred on 11 September at 22:37 UTC with an estimated $M_L$ of 3.1, after which proximal operations were suspended in line with the traffic light protocol introduced

for the KSMMA (BC Oil and Gas Commission, 2018). 73 precursory events occurred over approximately 4 hours prior to the $M_L$ 3.1 event, with events located within a small spatial extent ($\sim$300 m x 150 m), probably directly related to ongoing operations in the area due to the correlation in space and time of events and fluid injection. Events within this precursory sequence had magnitudes between $M_L$ 0.2 and $M_L$ 2.6, and were all located at depths of approximately 2.05 km. Moment tensor results for the $M_L$ 3.1 event suggest a focal mechanism dominated by strike-slip (Salvage et al., 2021).

### 3.2 Evidence of reduction in seismic noise: 2020

A clear reduction in the number of seismic events was observed during the lockdown period from April to August 2020 in the KSMMA (Fig. 3(b); grey background). Over the $\sim$ 4 months of relative quiescence only 389 events were detected using the EO network and available public stations in the area. For comparison, 344 events were detected on this network over a single week from 8 to 15 February when operations were fully underway. On average during this period, the magnitude of events were smaller than during time periods when activity was driven by ongoing operations.

A reduction in seismic noise and therefore ground motion is also evident in the KSMMA following the introduction of government restrictions and the subsequent economic downturn in March 2020 (Fig. 4). Unfortunately, the most central seismic stations in the EO array were not installed until immediately before and immediately after the end of lockdown (March and May, respectively), and therefore could not be used to analyze the long term changes in seismic noise. We chose station KSM08, located in the east of KSMMA, approximately 14 km due north of the settlement of Rolla, due to the long, uninterrupted seismic data recorded at this station, as well as its proximity to recent dense clusters of ongoing seismicity (Fig. 2). The recent seismicity close to KSM08 suggests that a number of wells in the vicinity were active prior to the economic downturn related to the COVID-19 pandemic and government imposed lockdowns in March 2020. Heightened seismic ground motion is evident at KSM08 through January to March, as operations are ongoing (Fig. 4). A significant decrease in average seismic ground motion is observed following the government restrictions in late March 2020, with the average displacement sitting well below the daytime mean calculated prior to lockdown. As restrictions ease, we see a large increase in ground motion following the reopening of businesses in May 2020, although this once again tails off through June and July. The decrease in average ground motion in the late summer is observed at all of the stations in the network, to varying degrees, and is believed to be related to the downturn in the industry (as a result of the COVID-19 pandemic) during which operations were suspended by companies in response to the COVID-19 pandemic (S. Venables, British Columbia Oil and Gas Commission, Pers. Comm., January 2021). The re-introduction of operations in August is clear from an increase in ground displacement and seismic noise, which has remained elevated ever since (although not as high as pre-lockdown levels).

### 3.3 Latent Seismicity during relative quiescence: 2020

Seismicity occurring during the period of quiescence from April to August 2020 within the KSMMA exhibit a number of characteristics indicative that it is a (latent) consequence of previous operations in the area. Figure 5 shows the temporal and spatial evolution of seismicity during this period. Firstly, perhaps unsurprisingly, seismicity does not occur in a distinct temporal

pattern that exhibits clustering (Fig. 5(a)). A small number of events (∼5) occur each day throughout the 4 month period. Event magnitudes also reveal no discerning patterns with time, with all events registering $M_L$ -0.6 to $M_L$ 1.2. Furthermore, the frequency index (FI) suggests no temporal patterns during the period of relative quiescence. The FI is a proxy for the spectral content of each waveform based upon the ratio of energy in low and high frequency windows (Buurman and West, 2010), calculated at a single station. We show our analysis for station KSM06 (Fig. 2) due to its proximity to the majority of the ongoing seismicity during this period of relative quiescence, although all stations within the KSM network suggested no temporal patterns in the FI. A negative FI means the waveform is dominated by low frequency energy (in this case 1 - 40 Hz); a positive FI demonstrates a majority of energy in the high frequency band (40.1 - 80 Hz). Overall, the seismicity detected during the period of relative quiescence within the KSMMA shows no discerning temporal characteristics.

Spatially, seismicity detected during the COVID-19 lockdown period exhibits characteristics that are similar to the previously detected seismicity in the KSMMA (Fig. 5(b)). Most events occur in a corridor orientated NW-SE, similar to the spatial distribution of seismicity prior to lockdown. Some spatial clustering is evident (e.g. in May in the south (shown in yellow)), but given the limited number of events this is difficult to determine with certainty. Most events during the quiescence period occur at focal depths of ∼0-4 km, which is similar to events prior to lockdown within the KSMMA, if potentially slightly shallower. Target formations for hydraulic fracturing within the KSMMA (Upper and Lower Montney) typically sit between 2 km and 2.5 km (total vertical depth), with salt-water disposal (SWD) injecting at shallower depths (M. Gaucher, British Columbia Oil and Gas Commission, Pers. Comm, November 2020). This suggests that events detected during the quiescence were generated in formations similar to those that occur when active hydraulic fracturing and SWD was ongoing.

## 4    Discussion

### 4.1    Characteristics of Observed Seismicity

Seismicity generated during this period of quiescence appears to share many characteristics with seismicity generated during hydraulic fracturing operations within the KSMMA. Although low in number, the event rate per day remains fairly constant throughout the ∼4 month period of limited hydraulic fracturing operations, with no apparent temporal decay (Fig. 5(a)). This contrasts the "usual" pattern of seismicity during active hydraulic fracturing operations, which is highly temporally (and spatially) clustered around the wells operating (Fig. 3) (e.g. Skoumal et al., 2015). Figure 3(b) also suggests no change in the recorded magnitudes of events pre- and during lockdown. Furthermore, no discernible changes in the FI are observed during the lockdown period (Fig. 5(a)). In volcanic environments (for which the FI was developed), a low FI is thought to be indicative of the presence of fluid moving through the system directly influencing the generation of seismicity (e.g. Lahr et al., 1994; Chouet, 1996; Salvage et al., 2018). However, in volcanic environments, low FI values typically contain energy of ≤5 Hz, with high FI values focused up to 15 Hz. The waveforms observed here contain much higher frequency energy than this, which is a common trait of seismicity generated in hydraulic fracturing environments (e.g. Eaton, 2018). Zoback et al. (2012) have suggested that

events dominated by low frequency energy in hydraulic fracturing environments may be indicative of slow-slip or aseismic deformation. Therefore, the low frequency nature of the events identified here may be indicative of either aseismic deformation or the presence of fluids within the system, however discerning between the two processes and their relative importance is still an area of active research.

The estimated magnitude of completeness (Mc) during the lockdown period is $\sim$0.4 (Fig. 6). We calculate Mc using the maximum-curvature method of Wiemer and Wyss (2000). Given the reduction in noise during the period of quiescence (Fig. 4), it is perhaps no surprise that a low Mc is identified. However, the Mc for 2020 (Mc$\sim$0.4, Fig. 3(b)) cannot directly be compared to the Mc for 2018 (Mc$\sim$1.0, Fig. 3(b)) since different methods were used in their magnitude calculations (Hutton and Boore (1987) vs. Babaie-Mahani and Kao (2020)). The lowering of Mc between these years is fundamentally related to an

increase in the monitoring capacity and detectability of small seismic events within the KSMMA (Salvage et al., 2021). The estimated $b$-value (Gutenberg and Richter, 1944) of 1.96 (Fig. 6, calculated using a least-squares linear regression for events with magnitudes $\geq$Mc) is similar to $b$-values estimated from seismicity associated with hydraulic fracturing experiments in western Canada, suggesting an abundance of lower magnitude events (Igonin et al., 2018; Eaton et al., 2014). The fact that no large magnitude events were detected during the period of quiescence (no $\geq M_L$ 1.5) is directly influencing the estimated

$b$-value in this case. Interestingly, higher $b$-values (as in this case) have typically been attributed to seismicity generated in normal faulting regimes (Schorlemmer et al., 2005; Amini and Eberhardt, 2019). The KSMMA is strongly influenced by the Fort St. John Graben complex, an asymmetrical half graben that has also undergone significant strike-slip and rotational movement upon reactivation of the basement faults in the area (Barclay et al., 1990), which may also be directly influencing the estimated $b$-value. Furthermore, in hydraulic fracturing environments, $b$-values of >2 have been associated with the stimulation

of natural fractures at depth, with smaller $b$-values associating with large-scale tectonic faults (Wessels et al., 2011; Eaton and Maghsoudi, 2015). In our case, this would suggest that the seismicity being generated may be related to both complex natural fracture systems and large scale faults in the area, since the $b$-value lies close to 2.

Seismicity during the quiescence appears to be spatially concurrent with previous seismicity in the area (Figs. 5(b) and 2).

However, there appears to be very little correlation between the spatial extent of seismicity and the most recent hydraulic fracturing activity in the area (active in March 2020 prior to lockdown; green sqaures, Fig. 5(b)). Seismicity during the quiescence appears in two planar elongated features, extending in a NW-SE direction, with lengths of up to 30 km (eastern segment), if assumed to be one feature. These features are not coincident with any known large-scale faults in the area (e.g. Furlong et al., 2020). Seismicity recorded during this period of quiescence is generally located at a similar depth to the target

formations of the Montney ($\sim$2 km), as well as in the formations above. This suggests hydraulically connected pathways above the injection zone, perhaps within mechanically stronger lithologies, as has been previously suggested by Eyre et al. (2019b) in the Fox Creek region of Alberta (another area undergoing intensive hydraulic fracturing operations).

The generation of induced seismicity has often been successfully correlated to a number of injection parameters, including the injected volume of fluid (e.g. Yu et al., 2019; Ellsworth, 2013) and/or the pumping rate (e.g. Goebel et al., 2017). Temporally data are too sparse to draw conclusions as to whether any of these parameters directly influence the generation of latent seismicity within the KSMMA, although given that hydraulic fracturing operations during our period of interest were extremely limited, we know that almost all of this seismicity cannot be a direct response of this type of fluid injection. Only one well pad was operated during this period of quiescence, from 19 to 24 July (S. Venables, British Columbia Oil and Gas Commission, Pers. Comm., January 2021), and only 4 events have both temporal correlation (i.e. occurred during this time window) and spatial correlation (i.e. occurred within 5 km of this well pad). However, there is evidence that a small number of seismic events identified from April to August 2020 may be associated with SWD. Within the KSMMA, only 8 SWD wells were active in 2020, compared to hundreds of hydraulic fracturing wells. Of these, only one well was active during our period of investigation (Fig 5(b)). We infer that the seismicity occurring on 13 April 2020 (Fig. 5a, upper panel), where over 20 events were registered on the same day (significantly above the background rate of seismicity during this quiescence), may be due to SWD, accounting for $\sim$6% of detected seismicity during this period of quiescence. In this case, ongoing sustained SWD occurred $\sim$2 km away from these events. This offset is not unusual for SWD and associated seismicity; Schultz et al. (2014) found an offset of $\sim$3.5 km between SWD and associated seismicity in Alberta.

## 4.2 Estimation of Noise

PPSD is one of the most common methods used to characterize ambient seismic noise. However, the level of smoothing, the size of the data window used in analysis and the methodology itself may all influence the PPSD calculation and distort features of interest (Anthony et al., 2020). Smoothing is primarily undertaken in order to reduce the uncertainty associated with the PPSD estimates, and means that short spikes in noise (e.g. due to wind gusts or seismic activity) do not dominate the spectrum. In our case, the reduction in ground motion is much easier to determine from the average of the PPSD rather than individual estimates (Figs. 1 and 4, green vs. grey lines). Although we use a period smoothing of 0.025 octaves, this is likely to provide adequate spectral resolution of spectral peaks, as shown by Anthony et al. (2020) and therefore impacts our results minimally. We also use a window of 30 minutes (overlapping by 50%) to try to reduce spectral leakage and variance when calculating the PPSD.

Earthquakes, and other transient signals, are likely to impact the estimation of ambient noise by generating large spikes in the data. However, the removal of seismicity from datasets is generally accepted as not necessary since they are low-probability occurrences within generally high-probability ambient seismic noise (McNamara and Buland, 2004). Only teleseismic earthquakes appear to have any real affect upon PPSD calculations (Anthony et al., 2020). A number of teleseismic events were detected during the period of quiescence analysis (e.g. $M_w$ 7.8 event on 20 July 2020, 99 km off the coast of Alaska), that may influence our calculation of PPSD. However, since we see no peak in the average ground motion at these times (e.g. no substantial peak in July 2020, Fig. 4), we suggest that teleseismic events are not majorly influencing our results.

One signal that does clearly influence our PPSD results in Figs. 1 and 4 is wind and other meteorological phenomena. Poor weather reported in the KSMMA, with wind gusts exceeding 80 km/hour at times (Government of Canada, 2020), were observed at the end of April during an otherwise quiet time period when there were limited hydraulic fracturing operations in KSMMA and limited movement of people due to lockdown measures. Since the noise generated from wind gusts penetrates a wide frequency band, we are unable to completely filter it out. Using a filter between 4 and 14 Hz tries to eliminate some of these transient signals mostly associated with meteorological and oceanic conditions.

### 4.3 Generation of Latent Seismicity

The suspension of almost all operations within the KSMMA in the summer of 2020 allows us a unique insight into seismicity that cannot be directly correlated with injection, which has always been the inferred triggering mechanism for most (if not all) of the seismicity within the KSMMA. The characteristics of the seismicity generated during this period suggest no fundamental differences in terms of temporal or spatial patterns or magnitudes to previous seismicity within the KSMMA that can be correlated with injection. In fact, many of the characteristics appear to be equivalent to events detected prior to lockdown. Prior to the development of the Montney play, low magnitude natural seismicity within the KSMMA was undetectable, given the limitations of the available seismic networks in place. The Canadian National Seismograph Network (CNSN) recorded 20 earthquakes ($M_L$ 2.5 - $M_L$ 4.3) from 1984 to 2008 within the KSMMA, which are assumed to be a mixture of natural events (Halchuk, 2009) and those associated with early anthropogenic activities in the area. Three clusters of events in November to December 1984 ($M_L$ 2.2 - $M_L$ 2.8), January to February 1992 ($M_L$ 2.5 - $M_L$ 3.5), and December 1992 to January 1993 ($M_L$ 2.5 - $M_L$ 4.1), have been associated with oil extraction and fluid injection (water) in the Eagle West and Eagle fields, just north of Fort St. John (Horner et al., 1994). One additional natural event occurred with a significantly larger magnitude in March 1986 ($M_w$ 5.4) north-east of Prince George, British Columbia (Halchuk, 2009), still a significant distance away from the study area.

In order to investigate the likelihood that this detected seismicity during the period of quiescence is natural seismicity, we calculate the expected recurrence rates of seismicity within the KSMMA greater than $M_L$ 2.5 from historical data, which is the magnitude of completeness used for the determination of seismic hazard maps in Canada due to detection thresholds from the Canadian public seismic network. The total number of earthquakes detected by the national network from 1984 to 2008 within KSMMA was 20 (Halchuk, 2009), suggesting a recurrence interval of 0.83 events per year. It is not unsurprising that during the period of quiescence, no events greater than $M_L$ 2.5 were detected, given this calculated recurrence interval. Following the Gutenberg-Richter formula (Gutenberg and Richter, 1944), it stands that there should be a 100-fold increase in the event rate to estimate the number of events >$M_L$ 0.5 (assuming a $b$-value of 1.0), suggesting an event count of 83. We use a $b$-value of 1.0, rather than our calculated $b$-value of 1.96 (Fig. 6) since this is the expected $b$-value for an area dominated by natural seismicity (e.g. Frohlich and Davis, 1993; Godano et al., 2014). Therefore, a maximum of 21% of events detected during the period of relative quiescence might be attributable to natural seismicity, in addition to the 6% attributable to SWD and one fluid injection well. However, this leaves ∼65% (a further 8% may be attributable to dynamic triggering effects, see below) of

seismicity generated during this period of relative quiescence that is difficult to explain by either of these mechanisms, and we suggest is more likely produced as a remnant to previous operations, and therefore directly related to previous states of stress. With events being generated over 4 months since the almost complete cessation of operations, the state of stress at depth must be near-critical for an extended period of time in order to generate this "latent" seismicity.

It is widely reported that earthquakes can be generated by transient stress changes related to the passage of seismic waves (i.e. "dynamic triggering") (e.g. Wang et al., 2015; Van der Elst et al., 2013; Hill and Prejean, 2007). In some cases, this dynamic triggering can also be delayed by days or weeks following a teleseismic event, potentially related to the re-distribution of pore fluid from the passing seismic waves (Brodsky and Prejean, 2005) or through initial aseismic slip on faults triggering seismicity (Shelly et al., 2011). During the period of quiescence (28 March to 6 August 2020), 43 earthquakes of >M6 were reported by the United States Geological Survey (2020), that may have the potential to cause dynamic triggering. We follow the methodology set out by Wang et al. (2015), whereby we first select only the teleseismic events that generated an estimated peak ground velocity of greater than 0.2 cm/s at any station within the KSMMA, as defined by Lay and Wallace (1995), whereby:

$$logA_{20} = M - 1.66log_{10}\delta - 2,$$ (1)

and:

$$PGV \approx \frac{2\pi A_{20}}{T},$$ (2)

where $A_{20}$ is the peak waveform amplitude when filtered at 20s; $M$ is the magnitude; $\delta$ is the epicenter-station distance (in degrees); and $T$ is the surface wave period (assumed to be 20 s). This method identified 40 events from the original list of teleseismic events. We then calculated the $\beta$ statistic (Matthews and Reasenberg, 1988) by:

$$\beta(N_1, N_2, t_1, t_2) = \frac{N_2 - E(N_2)}{\sqrt{var(N_2)}},$$ (3)

which is a quantitative measure of the level of dynamic triggering, representing the standard deviation in the background seismicity rate after a remote event. $N_1$ and $N_2$ are the number of earthquakes detected before ($t_1$) and after ($t_2$) the remote event, respectively. Here, we take $t_1$ and $t_2$ to be 12 hours. $E(N_2) = N_1 x t_2 / t_1$ is the expected number of earthquakes after the main shock based on the background seismicity rate. If no earthquakes occur in $t_1$ (i.e. before the main shock), $N_1$ is set to 0.25 based on the equivalent range of the probability density function (Matthews and Reasenberg, 1988; Hill and Prejean, 2007). When $\beta \geq 2$, there is sufficient statistical evidence (at a 95% confidence level) that there is a significant increase in the seismic event rate following the remote event (Hill and Prejean, 2007).

We identify 7 remote earthquakes that generate a $\beta$ value $\geq 2$ (Fig. 8), including the largest event to have occurred in 2020 located 99 km SSE of Perryville, Alaska on 22 July at 06:12 UTC, with $M_w$ 7.8 (United States Geological Survey, 2020). The

increase in event count in the KSMMA following this remote event is difficult to determine unless the above statistical analysis is performed, as it is difficult to see an increase in seismicity when daily event counts are used. In some cases however, such as following the $M_w$ 6.1 event on 31 May 2020, 43 km W of Lampa, Peru, a significant increase in the number of events detected

in KSMMA is clear. Our analysis therefore suggests that a maximum of 8% of the seismicity detected during this period of relative quiescence may be attributed to dynamic triggering, in particular the events on 31 May 2020. This rudimentary calculation does not take into account the potential spatial migration of events in response to the teleseismic event; only that if a significant increase in event count following the teleseismic event in the next 12 hours is observed, all events within this 12 hour window are determined to be related to dynamic triggering. However ~65% of the detected seismicity cannot be attributed

to primary activation mechanisms such as this, and therefore in our opinion are the result of "latent" ongoing processes.

Understanding how the stress field has evolved in time and space within the KSMMA, in particular with respect to the development of unconventional oil and gas operations in the area, is an extensive ongoing research topic. Very limited data regarding the state of stress prior to unconventional resource development in the area is available, since natural seismicity was

370 almost non-existent (e.g. Lamontagne et al., 2008) and details of such seismicity are often incomplete with large associated errors (e.g. Halchuk, 2009). Furthermore, the sparse nature of regional recording systems prior to unconventional development means that detailed understanding of the source characteristics of identified seismic events, which may allow insight into the state of stress, is not possible. Farahbod et al. (2015) investigated the changes in background regional seismicity within the Horn River Basin, British Columbia before and after hydraulic fracturing operations became prominent in the area. The Horn

River Basin is similar to the KSMMA in the fact that it is dominated by unconventional resource operations. Their study suggests that background seismicity dramatically increases following the introduction of hydraulic fracturing, correlating in time and space with ongoing operations, in a similar manner to what we have observed in KSMMA. However, their study fails to detail how the seismicity responds when hydraulic fracturing operations stop completely and whether there is a long-term change in the overall background rate of seismicity. If found to be true, this would indicate that the state of stress has changed

at depth to allow for such an increase.

Further evidence for a true change in stress at depth may be observed through changes in the principal stress axes. The World Stress Map suggests an average SHmax orientation of ~40° in the area close to Fort St. John, taken from borehole breakouts (Heidbach et al., 2018). Babaie-Mahani et al. (2020) calculated SHmax from focal mechanism analysis of 66 events

with magnitudes between 1.5 and 4.6 within the KSMMA associated with hydraulic fracturing, and found the value to be ~22° to ~33°, significantly rotated relative to the estimate from the World Stress Map. This would suggest that there is complexity in the stress distribution within the KSMMA, particularly spatially, possibly related to the structural complexity in the area as a result of the Fort St. John Graben complex (Barclay et al., 1990). The dominance of this tectonic feature throughout the KSMMA influences the identified source mechanisms of seismicity, which are extremely varied within a small spatial extent,

suggesting a complex and heterogeneous stress regime (Babaie-Mahani et al., 2020; Salvage et al., 2020; Amini and Eberhardt, 2019; Berger, 2012). As a consequence, determining a permanent change in the principal stress axes before hydraulic fracturing

began, during the dominance of hydraulic fracturing operations, and after the investigated period of quiescence requires further investigation.

The analysis presented here provides a rare opportunity to study seismicity in a period of relative quiescence during a time when the cessation of hydraulic fracturing operations within the KSMMA is not temporally or spatially limited (for example, as is the case when a seismic event $\geq$M3 occurs) nor is it due to the depletion of the entire reservoir when operations cease indefinitely. Instead, operations were suspended for $\sim$4 months due to an economic downturn as a result of the COVID-19 pandemic, across a wide area, which has once again seen an uptake in activity since August 2020 when the market recovered

(Fig. 3(b)). We are interpreting this identified seismicity as "latent" (i.e. long-lived with no obvious direct primary activation source) and suggest that it may be a response to previous fluid injection in this area, despite its long-lived nature. This may well represent a new "normal" background rate of seismicity in this area, since pore pressures at depth are presumably higher (in particular within less permeable formations) due to previous injected fluid, thus reducing the effective normal stress at depth, or may be the result of fault weakening with time due to previous operations. However, until operations within the KSMMA

cease permanently we will be unable to accurately measure this new background rate of seismicity, or determine whether the rates of seismicity observed here continue for sustained time periods (i.e. years).

    The generation of seismicity in response to hydraulic fracturing is typically attributed to either fluid migration models, poroelastic phenomenon, or potentially aseismic slip (e.g. Bao and Eaton, 2016; Langenbruch and Zoback, 2016; Shapiro

and Dinske, 2009; Segall and Lu, 2015; Eaton, 2018; Goebel and Brodsky, 2018; Eyre et al., 2019a). In the fluid migration model, pore fluid pressures are significantly increased upon fluid injection reducing the effective normal stress within a fault zone, which is sufficient to trigger seismicity (e.g. Peña Castro et al., 2020; Bao and Eaton, 2016). Given the temporal and spatial correlation between seismicity and hydraulic fracturing operations within the KSMMA, this appears to be a likely cause of some seismicity. Under this model, the seismicity rate is usually observed to be proportional to the pore pressure,

and is assumed to track the injection rate (Langenbruch and Zoback, 2016). Consequently, a slow and steady decrease in the rate of seismicity over time would be expected to occur, as fluid pressure leaks into the surrounding formations (Eyre et al., 2020), before seismicity returns to the background (i.e. natural) rate. Since seismicity during the period of quiescence is long-lived, shows no decay and cannot be attributed to increased fluid injection, another process must be involved in its generation. Furthermore, if pore fluid pressure and relaxation as a direct consequence of immediate injected fluid was the trigger of the

seismicity during this period of quiescence, we would expect the seismicity to spatially migrate directly outwards from the most recently injected wells. We see no evidence of this (Fig. 5(b)), suggesting direct pore fluid migration cannot be held responsible for the triggering of this sequence.

Seismicity triggered by pore pressure diffusion can be estimated by determining the propagating pore pressure fluid front (r$_t$) related to the hydraulic diffusivity in a homogeneous isotropic saturated poroelastic medium (Shapiro and Dinske, 2009; Parotidis et al., 2003) by:

$$r_t = 4\pi D t, \tag{4}$$

where $D$ is the hydraulic diffusivity and $t$ is time since injection. If the triggering front ($r_t$) closely follows the maximum distance of seismicity through time, then pore pressure diffusion is thought to play a central role in the triggering of this seismicity (e.g. Shapiro and Dinske, 2009; Parotidis et al., 2003). Diffusivity ($D$) is generally assumed to range in the Earth's crust between 0.1 m$^2$/s and 10 m$^2$/s (Scholz, 2019), although in areas affected by hydraulic fracturing it is thought to generally be in the range of 0.1 m$^2$/s to 2 m$^2$/s (e.g. Goebel et al., 2017; Shapiro and Dinske, 2009; Parotidis et al., 2003). Yu et al. (2019) suggested similar diffusivity values determined from seismicity related to hydraulic fracturing in the Montney formation to the north-west of KSMMA, although others have speculated that much smaller diffusion values would be expected in shale formations (Eyre et al., 2020; Guglielmi et al., 2015b). Higher values of diffusivity in hydraulic fracturing scenarios are believed to be the result of faults and fractures at depth acting as fluid corridors (e.g. Riazi and Eaton, 2020; Caine et al., 1996). However, the seismicity generated in the KSMMA during the period of quiescence shows no coherence with a triggering front from the most recently active injection wells (Fig. 7), suggesting that pore pressure diffusion is not the dominant mechanism responsible for triggering these earthquakes.

Other models proposed for the generation of seismicity in response to hydraulic fracturing suggest that both pore pressure and poroelastic effects are jointly responsible (e.g. Segall and Lu, 2015; Goebel and Brodsky, 2018). In these instances, the increased pore pressure due to injection is thought to load the surrounding rock matrix, altering the stress field, often at great distances from the original injection site, if the region is well hydraulically connected. Again, however, these models suggest that seismicity is generated as a response to injecting fluid into the Earth, which was occurring only on a very minor scale at the time of this seismicity. Given that the stress field is unlikely to be sustained at critical following the cessation of fluid injection, we would also expect a decay in seismicity with time (e.g. Utsu, 1961). We do not observe this. Alternatively, the trapping of fluids within a fault zone with only minor fluid migration along the fault, could result in slow changes to the effective stress due to changes in pore pressure (Sibson, 1992). In this scenario, seismicity should migrate spatially outwards from this fault zone as the effective stress migrates. We also see no evidence of this spatial migration (Fig. 5(b)).

Recently, it has been suggested that aseismic slip may play an important role in the generation of seismicity associated with hydraulic fracturing at distances extending beyond the fluid-pressurized zones through the transmission of an elastic stress perturbation (e.g. Eyre et al., 2019a; Bhattacharya and Viesca, 2019; Guglielmi et al., 2015a; Cappa et al., 2018; Wei et al., 2015). For hydraulic fracturing regimes in Canada, Eyre et al. (2019a) suggested that distal unstable regions of a fault may

be loaded by aseismic slip that initiated due to an increase in pore pressure within a stable zone, leading to the generation of seismicity. Once slip is initiated, far-field intraplate stresses may repeatedly reload unstable regions of the fault, leading to relatively steady seismicity rates, which may continue for long periods of time. They suggest the driving stresses of such behaviour are most likely to be elevated pore pressures (as a result of ongoing hydraulic fracturing in the area) becoming trapped within fault zones due to low permeabilities within many formations, sustained by tectonic forces. Cappa et al. (2018) suggested that far-field aseismic deformation may also be sustained by fault weakening, which may accelerate with time. Fault weakening may be further enhanced due to degradation of slip surfaces from chemical and hydrothermal fluid-rock interactions, as suggested in geothermal reservoirs by Vavryčuk and Hrubcová (2017). Given that in the absence of the cessation of operations the detection of latent seismicity is extremely difficult, there are few examples of long-lived seismicity associated with hydraulic fracturing operations that may offer insight into the generation of such activity. One recent example comes from a long-lived seismic swarm in Alberta, where seismicity was observed over 10 months after injection ceased, and was interpreted as being driven primarily by aseismic slip (Eyre et al., 2020). The authors suggest that the steady seismicity rate over a number of months (swarm-like behaviour of seismicity, rather than a typical mainshock-aftershock sequence) and a lack of hypocenter migration cannot be easily explained by a fluid-migration model (which is often the most favoured model in hydraulic fracturing environments), and is instead better explained by an aseismic slip model (Eyre et al., 2019a). We favour this interpretation of aseismic slip playing an important role in the initiation of this seismicity since ongoing hydraulic fracturing operations are not required to generate ongoing seismicity (as is the case in the fluid-migration model); instead, the trapping of fluids that have been previously injected within fault zones may be enough to sustain the generation of seismicity. The latent seismicity identified here is persistent over ∼4 months, lacks evidence of hypocenter migration and shows no evidence of a mainshock-aftershock type sequence as the magnitude distribution remains constant (i.e. is swarm-like in its behaviour). In addition we see no evidence of a propagating pore pressure fluid front and detected events are dominated by lower frequencies, all of which appear to be characteristic of the aseismic slip model (Eyre et al., 2019a, 2020; Bhattacharya and Viesca, 2019; Guglielmi et al., 2015a). Geodetic methods (e.g. GPS, InSAR) may be able to measure such aseismic deformation (e.g. Shirzaei et al., 2013; Biggs and Wright, 2020; Gualandi et al., 2017), however at this time no data of this nature is available for the KSMMA.

## 5 Conclusions

Seismicity generated in the KSMMA has been attributed to oil and gas recovery since production began in the area, primarily due to its temporal and spatial correlation to operations. However, during the COVID-19 pandemic in the summer of 2020, almost all hydraulic fracturing operations in the KSMMA were suspended. Despite this pause in industrial activity, 389 seismic events were recorded by our seismic network. These events occurred within the spatial extent of previous events in the area (a corridor orientated NW-SE), and had similar magnitudes to previously recorded seismicity (∼$M_L$ -1 to $M_L$ 1.2). The magnitude of completeness (Mc∼0.4) suggests that small events during this period of quiescence could be detected. The *b*-value of detected seismicity (∼1.96) is similar to previous estimates within areas dominated by hydraulic fracturing. Unlike

during active hydraulic fracturing operations, events showed no temporal clustering, but instead were generated in a persistent swarm-like manner over the $\sim$4 months of quiescence. No spatial correlation between the most recently active wells in the area and seismicity could be determined, however the fact that seismicity occurred at the depths of previous injection (i.e. within the target formations) suggests that the area is likely to be hydraulically linked.

Since there is no temporal or spatial evidence that these events are a direct consequence of the most recent hydraulic fracturing in the area (i.e. an aftershock sequence driven by pore pressure diffusion or poroelastic relaxation), and since the area is typically naturally relatively quiescent (a maximum of $21\%$ of the detected events may be attributable to natural seismicity rates), we suggest that most of these events may be an indirect response of the increased pore pressures at depth which is causing aseismic slip on already pressurized fault zones, or could be the result of fault weakening, as a result of previous fluid injection in the area. A number of events may be the result of dynamic triggering from remote events with $M_w$>6 (up to $\sim$8%), however this process cannot account for the majority of the seismicity observed ($\sim$65%). We suggest that the prior fluid injection in the area has altered the state of stress, and caused fluids to become trapped in fault and fracture zones at depth, in close proximity to the original injection points. This allows seismicity to be primarily generated by aseismic slip loading unstable regions of these pressurized zones at depth. Once slip has initiated, far-field tectonic stresses may repeatedly reload these unstable zones, leading to the relatively stable seismicity rate that is observed. The detected seismicity here during the period of quiescence may represent a new (heightened) background rate of seismicity in the KSMMA, however whether this is the case will only become apparent once operations cease permanently in the area.

*Code and data availability.* Continuous seismic data, station and associated metadata for the EO network is available through Incorporated Research Institutions for Seismology (IRIS) (http://ds.iris.edu/ds/nodes/dmc/) using Network Code EO, following a 91-day embargo period. The velocity model used for location analysis is available directly from the British Columbia Oil and Gas Commission. Seismic noise analysis (Figs. 1 and 4) was envisaged by Thomas Lecocq and Fred Massin, and can be found here: https://github.com/ThomasLecocq/SeismoRMS.

*Author contributions.* ROS contributed to the conceptualization, analyzed the data, wrote the manuscript and was primarily responsible for this research, with guidance, comments and revisions from DWE.

*Competing interests.* The authors declare that they have no conflict of interest.

*Acknowledgements.* The authors wish to thank Geoscience BC for funding this project. We also acknowledge our industry partners whose collaboration and funding enabled the installation of this network, including ARC resources, Canadian Natural Resources Ltd., and the Natural Sciences and Engineering Research Council of Canada. Further support was provided through the Microseismic Industry Consortium.

Nanometrics are gratefully acknowledged for their contribution, including the installation and maintenance of stations, and near real-time analysis of seismicity including the generation of the catalogue of seismic events. We would like to thank those at the Incorporated Research Institutions for Seismology (IRIS) for hosting the data and facilitating collaboration. We would especially like to thank Jay Hogan at Nanometrics who facilitated the successful upload of data to IRIS, as well as Stu Venables and Michelle Gaucher at the British Columbia Oil and Gas Commission for providing us with expert knowledge of the ongoing operations within the KSMMA from a regulatory perspective. We would like to thank Thomas H. A. Swinscoe (University of Calgary) for producing Fig. 2, which was produced using QGIS (https://qgis.org/en/site/about/index.html) and Thomas Lecocq (Royal Observatory Belgium) and Fred Massin (ETH Zurich) for codes that led to the development of Figs. 1 and 4. Figure 5(b) was produced using GMT v.6 (Wessel et al., 2019). Seismic analysis was carried out using the Obspy Python package (Beyreuther et al., 2010). Finally, we would like to thank C. Chamberlain, S. Crane, an anonymous reviewer and S. Hicks (Topical Editor) for their excellent comments and suggestions, which helped to much improve this manuscript, as well as all the Editors of this Special Edition of Solid Earth.

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

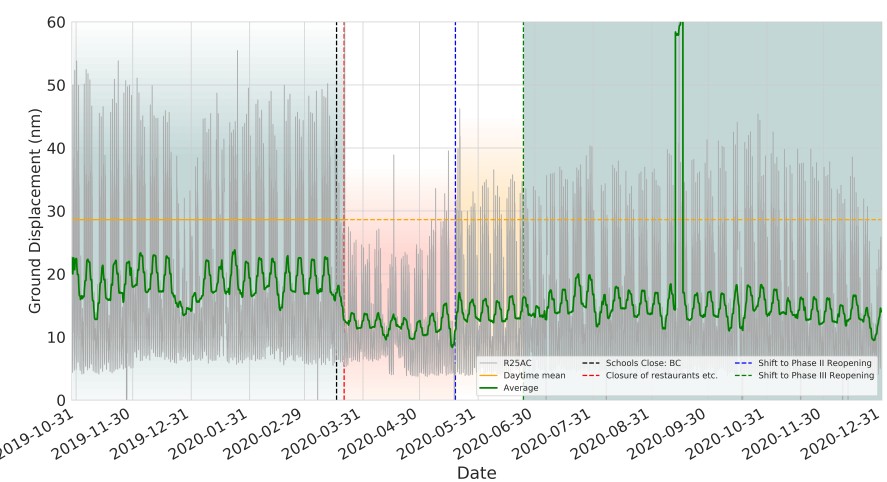

**Figure 1.** Filtered (4-14 Hz) ambient seismic noise displayed as displacement from station R25AC (vertical component) located in Vancouver, British Columbia. 30-minute average PPSD (dark grey), with rolling mean (window size = 92 hours) shown in green. The timing of different lockdown scenarios for British Columbia are shown as vertical dashed lines. Background colours represent different lockdown scenarios: Green represents before and after lockdown scenarios; Red is the first lockdown scenario with the closure of schools and restaurants; Yellow is the second lockdown scenario where some businesses re-opened. A clear reduction in the average ground motion is observed following initial lockdown conditions in March 2020. The large peak in noise in September is thought to be meteorological, rather than a sudden increase in anthropogenic activity. Figure courtesy of codes developed by Thomas Lecocq and Fred Massin.

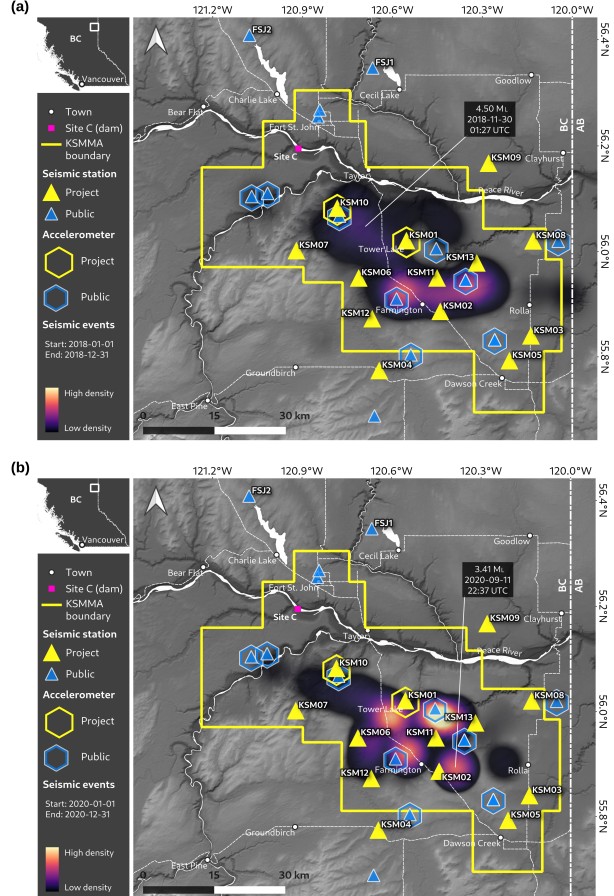

**Figure 2.** Spatial locations of seismicity concentrations within the KSMMA. Higher density of seismic events is indicated by brighter colours; lower density by darker colours; and no seismicity by grey. Densities displayed in (a) and (b) are directly comparable i.e. the absolute seismic density is the same in both subplots. The outline of the KSMMA boundary is shown in yellow; public seismic monitoring stations as blue triangles; the newly installed EO network as yellow triangles; and co-located accelerometers as hexagons. FSJ1 and FSJ2 are also part of the EO network but were installed in 2018. FSJ1 was decommissioned on 26 August 2020 but is shown for completeness as it was used in seismic analysis prior to this. The largest measured magnitude event within the KSMMA boundary for each year is marked. (a) Seismic events reported by NRCan between 1 January 2018 and 31 December 2018 (inclusive) (Visser et al., 2020). Note: although the new dense array was not installed at this time, it is shown on the map for reference. The largest event in 2018, occurring on 30 November, north of Tower Lake is shown ($M_L$ 4.5). (b) Seismic events recorded by the newly installed EO network (and incorporating data from public stations) from 1 January 2020 to 31 December 2020 (inclusive). The largest magnitude event, occurring on 11 September 2020 is indicated ($M_L$ 3.4). Figure courtesy of Thomas H. A. Swinscoe, University of Calgary, developed using QGIS (https://qgis.org/en/site/about/index.html).

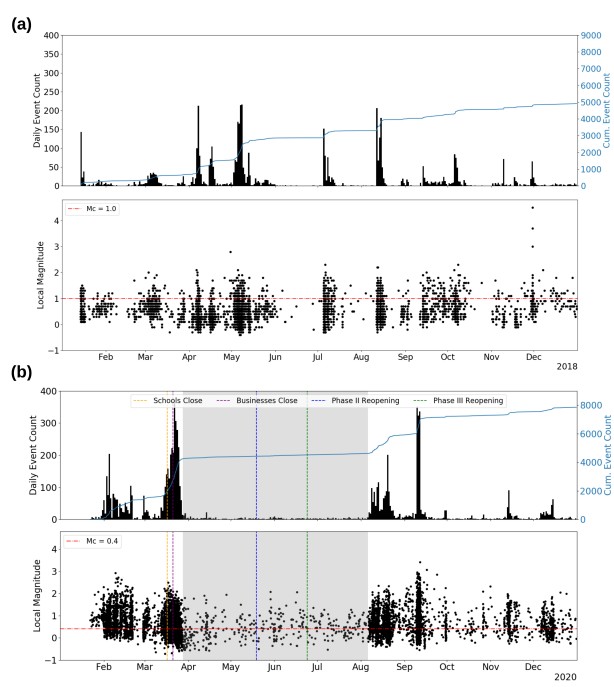

**Figure 3.** Temporal evolution of seismicity within the KSMMA (daily counts in black, cumulative counts in blue) for 2018 and 2020. Distinct temporal patterns can be observed in both years, in particular in the spring and autumn, associated with ongoing hydraulic fracturing operations in the area. (a) Seismic events reported by NRCan between 1 January 2018 and 31 December 2018 (inclusive) (Visser et al., 2020). Magnitudes were calculated using the $M_L$ formula of Hutton and Boore (1987). The magnitude of completeness (Mc) is reported as $M_L$ 1.0. (b) Seismic events recorded by the newly installed EO network (and incorporating data from public stations) from 1 January 2020 to 31 December 2020 (inclusive). Magnitudes were calculated using the specific local magnitude formula for the KSMMA introduced by Babaie-Mahani and Kao (2020). The magnitude of completeness (Mc) is estimated as $M_L$ 0.4. The timing of different lockdown scenarios affecting the KSMMA are shown as vertical dashed lines. The time period from April to August 2020 represents the period of relative quiescence analyzed in this paper (grey background) and is shown in more detail in Fig. 5. The time lag for seismicity build up after the Phase III reopening reflects the time required for operations in the area to be restarted.

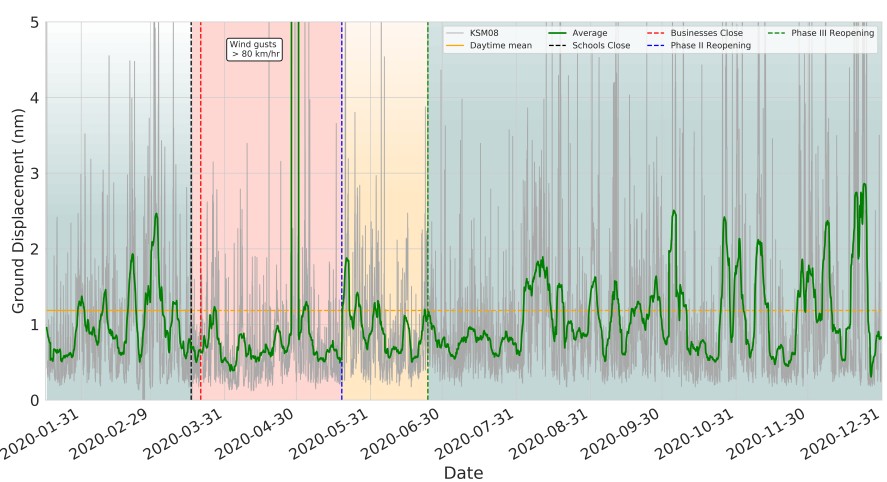

**Figure 4.** Filtered (4-14 Hz) ambient seismic noise displayed as displacement from station KSM08 (vertical component) located within the KSMMA. 30-minute average PPSD (dark grey), with rolling mean (window size = 92 hours) shown in green. The timing of different lockdown scenarios for British Columbia are shown as vertical dashed lines. Background colours represent different lockdown scenarios: Green represents before and after lockdown scenarios; Red is the first lockdown scenario with the closure of schools and restaurants; Yellow is the second lockdown scenario where some businesses re-opened. A clear reduction in the average ground motion is observed following initial lockdown conditions in March 2020, and significant increases in average ground motion as lockdown measures are rescinded. Figure courtesy of codes developed by Thomas Lecocq and Fred Massin.

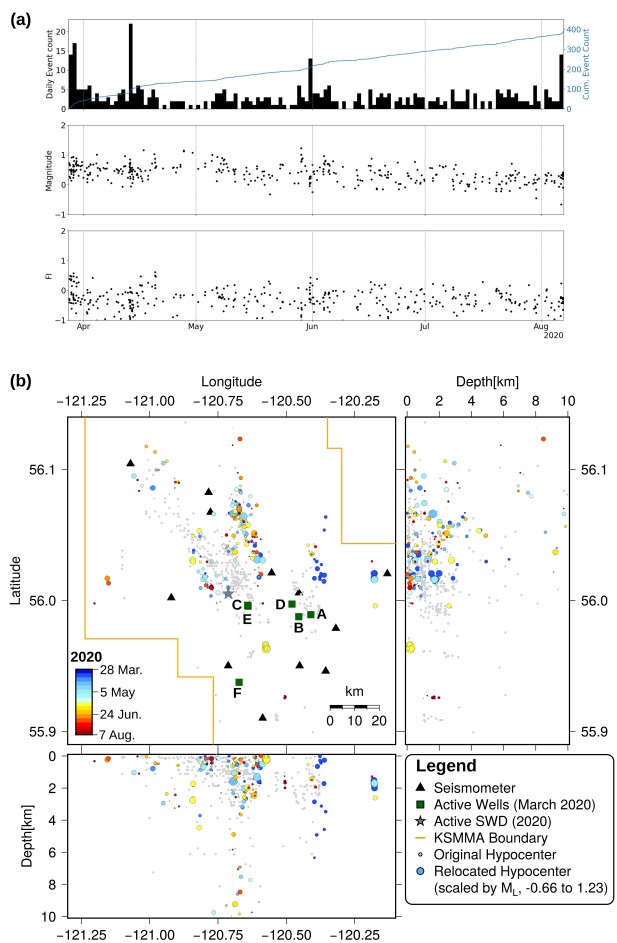

**Figure 5.** Temporal and spatial evolution of 389 events detected in the KSMMA during the cessation of almost all operations from April to August 2020. (a) *Upper:* Daily event count and cumulative event counts. *Middle:* $M_L$ determined using the formula of Babaie-Mahani and Kao (2020). *Lower:* Frequency Index (FI) detailing the ratio of high frequency energy to low frequency energy within each detected waveform at KSM06. (b) Spatial evolution of events coloured by time and scaled by magnitude. Active wells that initiated seismicity in the month prior to quiescence (March 2020) are shown as green squares labelled A (most recently active prior to lockdown i.e. late March 2020) to E (active in early March); one active SWD well is shown as the grey star. Figure generated using GMT v.6 (Wessel et al., 2019).

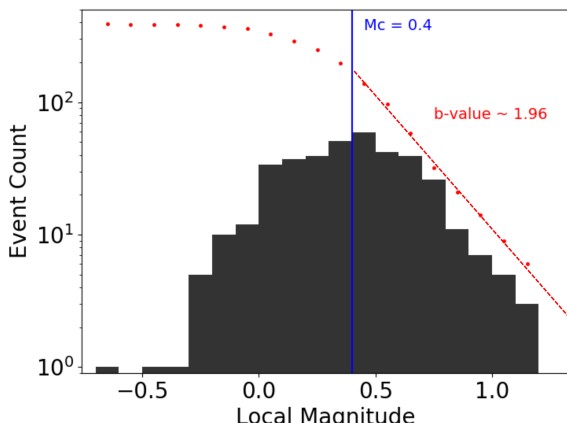

**Figure 6.** Frequency-Magnitude distribution of events (n=389) detected in the KSMMA during quiescence from 28 March to 6 August 2020. Event counts in magnitude bins of 0.1 are shown as black columns; the cumulative event value per bin is shown as a red dot. The magnitude of completeness (Mc) is 0.4, and the estimated $b$-value is 1.96 (Gutenberg and Richter, 1944). Local magnitudes were calculated following Babaie-Mahani and Kao (2020).

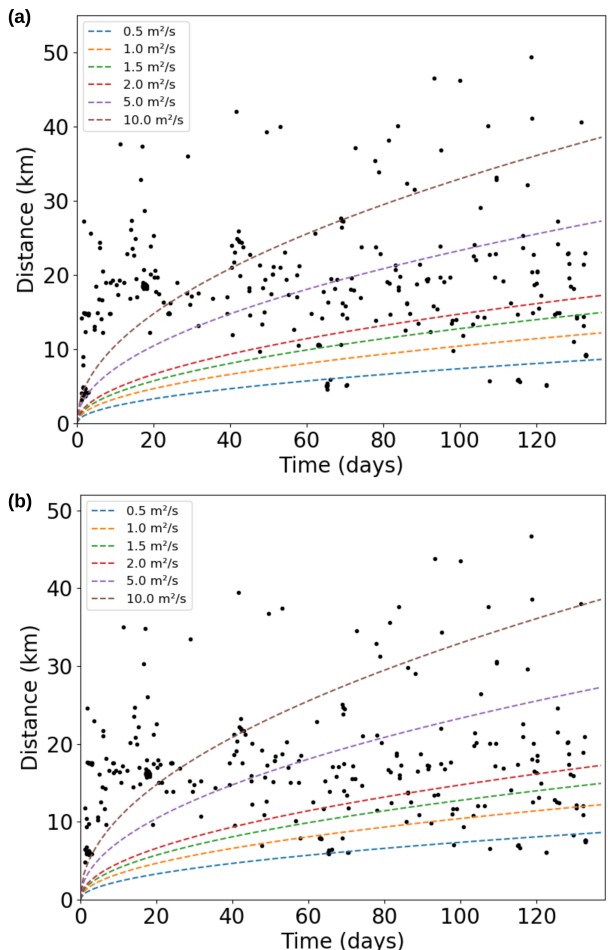

**Figure 7.** Time-distance plots of latent seismicity from 28 March to 6 August 2020. (a) Distance of events measured away from Well A (Fig. 5) and time zero taken as the last day of injection at this well prior to lockdown (27 March 2020). (b) Distance of events measured away from Well B (Fig. 5) and time zero taken as the last day of injection at this well prior to lockdown (27 March 2020). Representative diffusion curves associated with hydraulic fracturing and waste-water injection (Goebel et al., 2017; Shapiro and Dinske, 2009) are shown. The events during quiescence in the KSMMA cannot be successfully modelled using pore pressure diffusion suggesting it cannot be a primary mechanism for generating this seismicity.

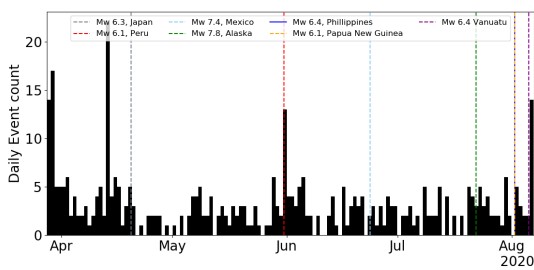

**Figure 8.** Teleseismic events (dotted vertical lines) that statistically (95% confidence level) generated dynamic triggering of seismicity within the KSMMA during a period of quiescence from April to August 2020. Daily event counts are shown as black bars. Statistical analysis ($\beta$ statistic following Matthews and Reasenberg (1988)) performed on the 12 hours prior to the teleseismic event, and 12 hours after.