# Peer review of "Unprecedented quiescence in resource development area allows detection of long-lived latent seismicity"

_Solid Earth, 2020_

## Referee Comment (RC1) · Calum J. Chamberlain (Referee) · 11 Jan 2021

**1   General Comment**

This paper documents the detection and analysis of earthquake activity within a normally-active region of hydraulic fracturing during the cessation of activity due to lockdowns associated with the COVID-19 pandemic. This study has an interesting and unique position in being able to assess changes in earthquake rate due to a change in hydraulic fracturing activity, alongside changes in earthquake detectability due to the reduction in background seismic noise. The authors find that seismicity during the

lockdown does not display the high-rate, temporally clustered sequences otherwise observed associated with reservoir stimulation, however, they do observe consistent seismicity within previously stimulated regions.

The authors provide good analysis of possible causes of this non-stimulated seismicity, including discussion of triggering from large regional earthquakes, the impact of pore-pressure, and longer-lasting fluid diffusion and poroelastic effects. The authors finally interpret this "latent" seismicity as being due to an altered stress-state within the previously stimulated regions due to trapped fluids, and infer that these earthquakes are driven by aseismic slip. I would like to see more discussion of the interplay between the purported stress-changes, and the strength of the fractures within the reservoir: I wonder if the change in stress is actually the dominant effect, or instead a reduction in fault strength due to prior fracturing would dominate the failure criterion? I'm also curious about how the purported high-pressures are sustained alongside the inter-connectedness of the fracture network?

Overall I think this is a good, well-written paper documenting an interesting case of reservoir stimulation shutdown. I think it might be relevant to point out the further uniqueness of this study in that most other shutdowns occur either after a large event, which would in itself alter the stress state, or when a reservoir is depleted. I have some specific comments below and some minor technical corrections to the manuscript.

**2 Specific Comments**

1. Is there any indication of long-lasting changes in the stress-field? e.g. do you observe changes in the stress-ratio or rotation in the principal stress axes associated with the initiation of reservoir stimulation, and is this sustained throughout the background seismicity? I imagine that the seismicity might be too sparse prior to the field becoming active to provide background state, but there may be

stress-field data from borehole-breakouts prior to stimulation? I'm also curious about the likely magnitude to stress-variation due to hydraulic fracturing.

2. Is there any other evidence of aseismic creep? I am not familiar with the paper by Eyre (2020), but I wonder if they found any characteristic temporal evolution of seismicity that they associated with aseismic slip? I'm also curious about why an aseismic driver is required? It is not generally assumed that background seismicity requires an aseismic driver, could this not just be the "new-normal" background seismicity after fractures were weakened due to hydraulic fracturing?

3. It would be great to have some well-stimulation data to confirm your suspicions in Line 125. I imagine that this is hard to come by, and if so, can you add a note around line 125 to say that well-data were not available.

4. Around line 267 the authors argue that upwards of 70

5. I am surprised that your magnitude of completeness appears to have gone up during the shutdown and I'm curious to hear why you think this is? It would also be useful to state what method was used to compute the magnitude of completeness.

**3  Technical Comments**

Below I have listed my technical comments starting with the relevant line number:

- 23: Change "on such" to "of such"

- 64: Change "Z-component" to "vertical-component"

[Figure]

- Lines 89-95: While the detection methods are not the key topic of this paper, and it only really matters that the detection method is consistent throughout the study, it would be good to have a little more detail and cite relevant papers - ideally citing another paper using the same methodology would be useful here.

- 134: What was the magnitude of completeness for these precursory events? Saying "A total" suggests absolute completeness.

- 175: Remove trailing "a" at the end of the line

- 218: "Data is" should be changed to "Data are" – it also isn't clear which data are being referred to here.

- 258: I suggest changing "non-existent" to "undetectable" given the limitations of the seismic networks available.

- 314: "Stress field would likely diminish": I suggest rephrasing this, it is hard to imagine the entire stress field diminishing, but there certainly might be a change in orientation and magnitude of principal stress axes. This statement could also do with a citation.

- 320: I don't know the paper cited here, but aseismic slip has long been associated with seismicity, so I assume that this paper talks specifically about aseismic slip in hydraulic fracturing environments: It would be good to make that clear in this statement.

- 365: Change "always" to "since Oil and Gas production started" or similar.

- Figure 2(b): The inset and key are not needed as far as I can tell because they repeat from 2(a) – happy to have them left in, but if you do, can you add in the white box on the inset map showing the main figure location – I saw this in 2(a), but not 2(b).

[Figure]

---

## Referee Comment (RC2) · Stephen Crane (Referee) · 12 Jan 2021

General Comments:

Overall, this well-written paper establishes the presence of "latent" seismicity in an area where resource development had stopped due to restrictions put in place to prevent the spread of COVID-19. This paper estimates that 70% of the earthquakes measured in the Kiskatinaw area, B.C., between April and August 2020, are due to aseismic slip from leftover fluids becoming trapped in the target formation, after the resource development in the area had been paused. The other 30% of seismicity is either background seismicity, which has been occurring before the resource development in the area, or

directly triggered from a teleseismic earthquake.

The authors establish the background of the Kiskatinaw Seismic Monitoring and Mitigation Area (KSMMA), the resource development occurring in the area, and the past and current seismic monitoring taking place. The unique conditions established here involve an expanded seismic network, yet a stoppage of hydraulic fracturing operations due to a global pandemic and not a large seismic event or a depletion of the reservoir as is the case in other areas. The seismic network allows for the detection of small earthquakes, in a region that experiences relatively low natural seismicity.

In this paper it is shown that the most of the detected earthquakes in the area between April and August 2020 are: not the result of direct injection, as there are no spatial or temporal clusters around wells and no active wells during this period; not the result of natural seismicity, as the previously measured seismicity rates are too small to account for the number of events detected; and not triggered by large teleseimic earthquakes. This leads the authors to conclude that previous fluid injections in the area altered the state of stress in a hydraulically linked formation generating aseismic slip loading on unstable zones.

This paper does a good job of testing the possibilities of causes to seismicity in an area after resource development has paused, where seismicity was uncommon beforehand. After testing the seismicity traits against those that are common from other causes of seismicity, it is determined in this area that the seismicity measured over this period can be labelled as "latent" seismicity. That is, the seismicity is caused by prior resource development in this region, but not directly related to active hydraulic fracturing or salt water disposal.

Specific Comments:

1. In Section 3.3 the FI index is discussed as way to describe whether fluids play a direct role in seismicity (negative FI) or not (positive FI). I understand that there are no temporal variations of FI, but was a consistently negative FI or consistently positive FI

found (Fig 5a)? And would that imply a direct role of fluids or not in the seismicity?

2. Lines 190-193: How is the magnitude of completeness measured? Is it spatially distributed over the KSMMA boundary based on the station density or calculated as the point where the catalogue deviates from the calculated b-value (as shown in Figure 6)? I think there should be a reference to Figure 6 here.

Technical Comments:

Line 10 "locate" should be "located"

Line 78-79: "In fact, ground displacement remained between 20 and 30 nm at station R25AC for the entirety of 2019." Should this supposed to be the average ground displacement that remained between 20 and 30 nm?

Line 99-101: This sentence needs a proper ending ie. "... previous seismicity (available directly from BCOGC), was used to determine the locations in NonLinLoc"

Line 111 and Figure 3 caption: "In both years" appears before the mention of 2018 and 2020. It would be helpful to give the specific years before referring to them as both years.

Line 130: "to present" could this now be replaced by a specific date or month?

Line 258: CNSN stands for Canadian National Seismograph Network.

Line 268-270: There are 3 "therefore" in 2 sentences.

Figure 3: I think it would be easier to compare if the two figures had the same x-axis, and the note mentioning that the 2020 data is complete only until October. Comparing the temporal patterns on different time scales seems difficult.

Figure 6. Shown but never referenced in the text.

---

## Referee Comment (RC3) · Anonymous Referee #3 · 17 Jan 2021

General comments:

This nicely written and well-structured article examines how seismicity rates during quiescent periods can be used to examine driving factors behind 'latent' induced seismicity after direct injection is stopped. The authors study seismicity in the Kiskatinaw area of British Columbia before, during, and after the 2020 COVID-19 government-imposed shutdown to investigate how the seismicity behaves when fluid injection operations in the region temporarily stop. They find evidence for "latent" seismicity during the quiescent period, of which 70% cannot be explained by direct injection, natural seismicity, or triggered earthquakes. After comparing the seismicity trends to those expected from

fluid migration models, pore pressure diffusion, poroelastic effects, and aseismic slip, the authors eliminate all but aseismic slip as the most likely driving factor to the latent seismicity. Throughout this process, the authors provide clear analysis and evidence to their arguments.

This paper is a unique and timely article with respect to the current pandemic and global lockdowns. The authors apply common and suitable seismological techniques to investigate earthquakes that would otherwise be hidden by direct injection seismicity. I think this article will be of interest to the audience of Solid Earth. Please find the specific and technical comments below.

Specific comments:

Line 28: This number (∼0.3%) has been updated in a later publication: "Ghofrani & Atkinson (2020), Activation Rate of Seismicity for Hydraulic Fracture Wells in the Western Canada Sedimentary Basin, BSSA". Regionally, they now estimate ∼0.8%.

Line 64: Could the authors also include why only the Z-component was used in the study, and not any horizontals.

Line 80 – Section 3: Please explain why you chose to compare to 2018 and not 2019 in this section. You compared the seismic noise levels to 2019 (lines 77-78), why not also the seismicity?

Line 80 – Section 3: Also, please include somewhere how you processed the time series, any band-pass filter applied? SNR?

Line 82-88: More details on the instruments would be good, e.g., sample rates? Are the newly installed stations specifically chosen in some way for certain magnitudes? Site conditions (same for all stations, or different)?

Line 110 – Figure 2: Is the absolute seismicity density the same for the two subplots? I.e. is high density for (a) = high density for (b)? To compare the two periods, it would be good if the colours refer to similar seismicity densities. If not, please make this clear

in the caption. Also, is the seismicity used the same as in Figure 3?

Line 113 – Figure 3: Could the authors include the magnitude of completeness for the two different datasets in the figure (to make it easier to compare).

Line 149-150: Could the authors include more details on where the KSM08 station is located? Is it far away from cities/towns? Near any wells?

Line 153-155 – Figure 4: Any ideas as to why the seismic noise level is low in July at KSM08? Looks to be down at the same levels as during April. Is this a trend seen at more stations than just KSM08?

Line 165: For the FI value, do you compensate for high-frequency attenuation in some way? You mention that you use one station for all events, won't the low-to-high ratio be different depending on how much high-frequency energy has been attenuated? I.e., events from larger distances have less high-frequency content due to more attenuation than the closer events.

Line 188-189: Visually, Figure 3b appears to have a slightly decreasing magnitude trend with time. The cloud is around ML 0.0 to 1.0 in June, and ML -0.5 to 0.5 in August. Have you looked into this? Any tests done to find trends?

Line 191-204: It would be interesting to see a second plot from before lockdown and a third after seismicity picked back up again. How does the b-value change between the three periods?

Line 356: Would it be possible for the authors to instead plot each event as a circle (e.g., based on magnitude as in previous plots) so that they can highlight the events they identified as triggered by a remote event?

Technical comments:

General technical: figure font size was quite small and needs to be increased.

Line 7-11: Regarding the three sentences: the authors write that "general characteristics" are similar between active and shutdown periods, but then go on to state two reasons they are different (magnitude + temporal clusters) and only one reason they are similar (spatially). This makes it seem like they are more different than similar. Perhaps rephrase first sentence.

Line 45: Sentence structure is off. "We call this latent seismicity i.e. seismicity..."

Line 62: Comma missing: "Following the methodology of Lecocq et al. (2020) we compute the..."

Line 72 – Figure 1: Please also explain what the vertical highlighted (yellowish) periods are in the figure caption (occurring before vertical dashed red line). Also, text is very small.

Line 75-77: Sentence is confusing, consider rearranging: "Following the reopening of some businesses in May and June 2020, the increase in noise is interpreted as the increased movement of people, although it remains lower than pre-pandemic levels." Or something similar.

Line 87: Why not reference Figure 2 here for the station configuration?

Line 99-101: Sentence doesn't make sense.

Line 104-107: In Figure 3 caption, you reference Hutton and Boore (1987) as the origin of your ML calculations. This is not who you reference in the text.

Line 112: "ML 3-4+" doesn't really make sense, either it's ML 3-4, or ML 3+.

Line 113 – Figure 3: Please use the same y-axis limits on the a) and b) plots since we're supposed to compare them.

Line 114-115: You only state in the Figure 3 caption that the seismicity increase in 2020 March, August, and September are due to hydraulic fracturing operations. Please include this in the text instead of in the figure caption.

Line 149-150: Comma placement in this sentence is a bit off.

Line 156: "a pre-lockdown levels" is grammatically incorrect.

Line 161 – Figure 5: Here it says you use the Babaie-Mahani & Kao (2020) formula to compute ML. Same or different to the one in Figure 3?

Line 177-178: "2000 m and 2500 m" please switch to km to stay consistent with previous sentence.

Figure 6: is not referenced in the text.

Line 247-249: Strange sentence structure with the commas and parentheses.

Line 299: 'it' is missing: "in areas affected by hydraulic fracturing it is thought to. . ."

---

## Author Comment (AC1) · 12 Feb 2021

*All line numbers refer to line numbers in the updated "clean" manuscript i.e. that without the track changes.*

**1    General Comments**

**This paper documents the detection and analysis of earthquake activity within a normally-active region of hydraulic fracturing during the cessation of activity**

[Figure]

**due to lockdowns associated with the COVID-19 pandemic. This study has an interesting and unique position in being able to assess changes in earthquake rate due to a changein hydraulic fracturing activity, alongside changes in earthquake detectability due tothe reduction in background seismic noise. The authors find that seismicity during the lockdown does not display the high-rate, temporally clustered sequences otherwise observed associated with reservoir stimulation, however, they do observe consistent seismicity within previously stimulated regions.**

**The authors provide good analysis of possible causes of this non-stimulated seismicity, including discussion of triggering from large regional earthquakes, the impact of pore-pressure, and longer-lasting fluid diffusion and poroelastic effects. The authors finally interpret this "latent" seismicity as being due to an altered stress-state within the previously stimulated regions due to trapped fluids, and infer that these earthquakes are driven by aseismic slip. I would like to see more discussion of the interplay between the purported stress-changes, and the strength of the fractures within the reservoir: I wonder if the change in stress is actually the dominant effect, or instead a reduction in fault strength due to prior fracturing would dominate the failure criterion? I'm also curious about how the purported high-pressures are sustained alongside the inter-connectedness of the fracture network?**

*These are excellent points, and we have tried to address them in the discussion section of the manuscript (L 400-405; L 451-463). We agree that a change in stress or a reduction in fault strength may be responsible for generating this seismicity, which we did not really discuss before, and so we have added fault weakening into our discussion as a possible cause. We have also added more discussion around how aseismic slip may enable the generation of seismicity beyond the pressurized fluid*

*front (i.e. at large distances from the injection point), as has been hypothesised by a number of authors and have included a number of references (e.g. Eyre, 2019; Wei et al., 2015; Cappa et al., 2018). We think that far-field (tectonic) stresses may play a role in sustaining such slip, as well as direct interaction with a pressurized fault patch over shorter distances.*

**Overall I think this is a good, well-written paper documenting an interesting case of reservoir stimulation shutdown. I think it might be relevant to point out the further uniqueness of this study in that most other shutdowns occur either after a large event, which would in itself alter the stress state, or when a reservoir is depleted. I have some specific comments below and some minor technical corrections to the manuscript.**

*Thank you for your comments. We have tried to emphasize the more unique aspect of this study throughout the text, and in particular in L 59-64 and L 394-399, as you suggest.*

**2   Specific Comments**

1. **Is there any indication of long-lasting changes in the stress-field? e.g. do you observe changes in the stress-ratio or rotation in the principal stress axes associated with the initiation of reservoir stimulation, and is this sustained throughout the background seismicity? I imagine that the seismicity might be too sparse prior to the field becoming active to provide background state, but there may be stress-field data from borehole-breakouts prior to stimulation? I'm also curious about the likely magnitude to stress-variation due to hydraulic fracturing.**

   *You are quite correct that the seismicity is too sparse prior to the KSMMA be-*

*coming active to really detail this. Source characteristics of the events identified as "background" seismicity" are mostly not accounted for. We did find some information regarding borehole breakouts, and a recent study by Babaie-Mahani et al. (2020) which details recent (2018-2019) stresses (from focal mechanism analysis), however a comparison of before and after is not possible due to the limited data. We have added a number of paragraphs into the discussion for this (L 366-393), however at this time we are unable to categorically give an answer to this question as it represents a large area of research that is yet to be undertaken.*

2. **Is there any other evidence of aseismic creep? I am not familiar with the paper by Eyre (2020), but I wonder if they found any characteristic temporal evolution of seismicity that they associated with aseismic slip? I'm also curious about why an aseismic driver is required? It is not generally assumed that background seismicity requires an aseismic driver, could this not just be the "new-normal" background seismicity after fractures were weakened due to hydraulic fracturing?**

*We have now included more references to aseismic slip in hydraulic fracturing environments, and tried to point out the characteristics that we observe in our seismicity that we believe are indicative of this mechanism (L 473-477). This includes the long-lived persistant nature of the seismicity (swarm-like activity rather than a typical mainshock-aftershock sequence), the lack of hypocenter migration (away from a point of injection), as well as dominance of low frequency energy in the waveforms.*

*It is possible that the seismicity detected reflects a new normal in terms of background seismicity, with event counts now greater than they were prior to operations in this area, and we have added this comment in to the manuscript*

*(L 400-405). We were trying to suggest that the generation of this seismicity (whether it is latent and a direct consequence of the recent operations, or whether it is the new background rate) could be generated by aseismic slip (as a result of tectonic forces), or through other mechanisms such as fracture weakening. We have tried to make this more clear in the manuscript by softening our language to advocate that aseismic slip is a proposed mechanism for this seismicity generation, rather than the only mechanism.*

3. **It would be great to have some well-stimulation data to confirm your suspicions in Line 125. I imagine that this is hard to come by, and if so, can you add a note around line 125 to say that well-data were not available.**

*As you note, this information is proprietary between the regulators and the individual companies operating in this area, and as such we cannot comment further on this as we do not have access to this information.*

4. **Around line 267 the authors argue that upwards of 70% of the earthquakes "cannot be explained by this mechanism [natural seismicity]". The argument is based on Gutenberg-Richter scaling, which is quite a weak argument for such a strong statement. Furthermore, the statement of scaling is based on a b-value of 1, which the authors do not find for this region - I'm curious about the reason for using a b-value of 1? Finally: it is hard to make such a strong statement, given that natural earthquake (de-clustered) distributions appear to be almost Poisson and can have variations in rate in time. I suggest more circumspect language in this statement rather than "cannot be explained".**

*We agree that the language here needs to be softened, and we have tried to do so throughout the text in relation to this point. We used a b-value of 1 since*

*this is the expected b-value for natural seismicity in North America (Frohlich and Davis, 1993; Godano et al., 2014), and as we were trying to determine whether any events could be deemed "natural", we chose this. We have added this explanation into the text (L 321-323). It is common to have much higher b-values associated with hydraulic fracturing experiments (as we noted in the text and found in our study), but the aim of this part of the discussion was to determine whether any/some/all of the seismicity may be natural. We have tried to make this clearer.*

5. **I am surprised that your magnitude of completeness appears to have gone up during the shutdown and I'm curious to hear why you think this is? It would also be useful to state what method was used to compute the magnitude of completeness.**

   *We have added the method we used to calculate the magnitude of completeness (maximum curvature method). We originally included this comment for completeness, however, upon reflection we can see that it is confusing and we have therefore removed the sentence regarding the Mc for the entire catalogue from this paper. We believe that the differing Mc is an artefact due to differing methods used in the magnitude calculations between Salvage et al., 2021 and this paper, meaning they are not directly comparable.*

**3  Technical Comments**

Below I have listed my technical comments starting with the relevant line number:

- **23: Change "on such" to "of such"**

  *This was a typo and has been changed.*

- **64: Change "Z-component" to "vertical-component"**

  *We have changed all references to Z-component to now read the vertical component.*

- **Lines 89-95: While the detection methods are not the key topic of this paper, and it only really matters that the detection method is consistent throughout the study, it would be good to have a little more detail and cite relevant papers - ideally citing another paper using the same methodology would be useful here.**

  *We have added a couple of sentences relating to the methodology employed here, as well as reference to a paper that uses it (L 106-114).*

- **134: What was the magnitude of completeness for these precursory events? Saying "A total" suggests absolute completeness.**

  *This was not our intended meaning. We have changed the sentence to reflect that we simply wanted to state the number of events in the precursory sequence. Due to the small number of events in this sequence, the Mc was calculated to be ~1.1.*

- **175: Remove trailing "a" at the end of the line**

  *This has been removed.*

- **218: "Data is" should be changed to "Data are" - it also isn't clear which data are being referred to here.**

  *We have changed this to be gramatically correct.*

- **258: I suggest changing "non-existent" to "undetectable" given the limitations of the seismic networks available.**

  *This has been changed.*

- **314: "Stress field would likely diminish": I suggest rephrasing this, it is hard to imagine the entire stress field diminishing, but there certainly might be a change in orientation and magnitude of principal stress axes. This statement could also do with a citation.**

  *We have changed this sentence to be more accurate (and realistic), and included a reference as you suggest (L 446-447).*

- **320: I don't know the paper cited here, but aseismic slip has long been associated with seismicity, so I assume that this paper talks specifically about aseismic slip in hydraulic fracturing environments: It would be good to make that clear in this statement.**

  *This has been changed.*

- **365: Change "always" to "since Oil and Gas production started" or similar.**

  *This has been changed.*

- **Figure 2(b): The inset and key are not needed as far as I can tell because they repeat from 2(a) - happy to have them left in, but if you do, can you add in the white box on the inset map showing the main figure location I saw this in 2(a), but not 2(b).**

  *Figure 2 has been updated to include all seismicity from 2020 (at the time of submission it only included seismicity to October). We have also increased the text size in the figure, and made sure the inset map includes the white box. We have left the legend in both plots.*

---

## Author Comment (AC2) · 12 Feb 2021

*All line numbers refer to line numbers in the updated "clean" manuscript i.e. that without the track changes.*

**1    General Comments**

**Overall, this well-written paper establishes the presence of "latent" seismicity in an area where resource development had stopped due to restrictions put in**

[Figure]

place to prevent the spread of COVID-19. This paper estimates that 70% of the earthquakes measured in the Kiskatinaw area, B.C., between April and August 2020, are due to aseismic slip from leftover fluids becoming trapped in the target formation, after the resource development in the area had been paused. The other 30% of seismicity is either background seismicity, which has been occurring before the resource development in the area, or directly triggered from a teleseismic earthquake.

The authors establish the background of the Kiskatinaw Seismic Monitoring and Mitigation Area (KSMMA), the resource development occurring in the area, and the past and current seismic monitoring taking place. The unique conditions established here involve an expanded seismic network, yet a stoppage of hydraulic fracturing operations due to a global pandemic and not a large seismic event or a depletion of the reservoir as is the case in other areas. The seismic network allows for the detection of small earthquakes, in a region that experiences relatively low natural seismicity.

In this paper it is shown that the most of the detected earthquakes in the area between April and August 2020 are: not the result of direct injection, as there are no spatial or temporal clusters around wells and no active wells during this period; not the result of natural seismicity, as the previously measured seismicity rates are too small to account for the number of events detected; and not triggered by large teleseimic earthquakes. This leads the authors to conclude that previous fluid injections in the area altered the state of stress in a hydraulically linked formation generating aseismic slip loading on unstable zones.

This paper does a good job of testing the possibilities of causes to seismicity

**in an area after resource development has paused, where seismicity was uncommon beforehand. After testing the seismicity traits against those that are common from other causes of seismicity, it is determined in this area that the seismicity measured over this period can be labelled as "latent" seismicity. That is, the seismicity is caused by prior resource development in this region, but not directly related to active hydraulic fracturing or salt water disposal.**

*Thank you for this nice summary and positive comments.*

**2 Specific Comments**

1. **In Section 3.3 the FI index is discussed as way to describe whether fluids play a direct role in seismicity (negative FI) or not (positive FI). I understand that there are no temporal variations of FI, but was a consistently negative FI or consistently positive FI found (Fig 5a)? And would that imply a direct role of fluids or not in the seismicity?**

   *We have added a couple of sentences in the discussion where we suggest that the FI is not a commonly used measure in hydraulic fracturing seismicity (it was developed for volcanic environments) and therefore we base the fact that larger amounts of low frequency energy within the waveform is indicative of the role of fluid in its generation upon research in the volcanological community (L 223-230).*

2. **Lines 190-193: How is the magnitude of completeness measured? Is it spatially distributed over the KSMMA boundary based on the station density or calculated as the point where the catalogue deviates from the calculated b-value (as shown in Figure 6)? I think there should be a reference to Figure 6 here.**

   *You are quite correct that we were missing a reference to Fig. 6 at this point. It has now been added. We have also added a comment about how we compute*

*the magnitude of completeness (maximum curvature method) and the $b$-value*
*(least-squares linear regression) for completeness (L 231-237).*

**3  Technical Comments**

- **Line 10 "locate" should be "located"**

  *This has been changed.*

- **Line 78-79: "In fact, ground displacement remained between 20 and 30 nm at station R25AC for the entirety of 2019." Should this supposed to be the average ground displacement that remained between 20 and 30 nm?**

  *This has been changed.*

- **Line 99-101: This sentence needs a proper ending ie. "... previous seismicity (available directly from BCOGC), was used to determine the locations in NonLinLoc"**

  *Another reviewer also mentioned this, so this sentence and the previous one have been re-written for clarity.*

- **Line 111 and Figure 3 caption: "In both years" appears before the mention of 2018 and 2020. It would be helpful to give the specific years before referring to them as both years.**

  *Changes in the figure caption and the text mean that this is no longer the case.*

- **Line 130: "to present" could this now be replaced by a specific date or month?**

  *We have now updated the text and the figures to reflect seismic data for the entirety of 2020.*

- **Line 258: CNSN stands for Canadian National Seismograph Network.**

  *Thank you for this clarification.*

- **Line 268-270: There are 3 "therefore" in 2 sentences.**

  *We have used a number of different synonyms now to avoid our use of "therefore".*

- **Figure 3: I think it would be easier to compare if the two figures had the same x-axis, and the note mentioning that the 2020 data is complete only until October. Comparing the temporal patterns on different time scales seems difficult.**

  *The figure has now been updated to represent data for the entirety of 2020, and so both subplots are now on the same x-axis.*

- **Figure 6. Shown but never referenced in the text.**

  *This was an oversight on our part and has now been corrected.*

---

## Author Comment (AC3) · 12 Feb 2021

[id=ROS]All line numbers refer to line numbers in the updated "clean" manuscript i.e. that without the track changes.

**1   General Comments**

**This nicely written and well-structured article examines how seismicity rates during quiescent periods can be used to examine driving factors behind "latent"**

[Figure]

induced seismicity after direct injection is stopped. The authors study seismicity in the Kiskatinaw area of British Columbia before, during, and after the 2020 COVID-19 government-imposed shutdown to investigate how the seismicity behaves when fluid injection operations in the region temporarily stop. They find evidence for "latent" seismicity during the quiescent period, of which 70% cannot be explained by direct injection, natural seismicity, or triggered earthquakes. After comparing the seismicity trends to those expected from fluid migration models, pore pressure diffusion, poroelastic effects, and aseismic slip, the authors eliminate all but aseismic slip as the most likely driving factor to the latent seismicity. Throughout this process, the authors provide clear analysis and evidence to their arguments.

This paper is a unique and timely article with respect to the current pandemic and global lockdowns. The authors apply common and suitable seismological techniques to investigate earthquakes that would otherwise be hidden by direct injection seismicity. I think this article will be of interest to the audience of Solid Earth. Please find the specific and technical comments below.

*Thank you for these comments - we are glad the reviewer found our article of interest.*

**2   Specific Comments**

1. **Line 28: This number ($\sim$0.3%) has been updated in a later publication: "Ghofrani $\&$ Atkinson (2020), Activation Rate of Seismicity for Hydraulic Fracture Wells in the Western Canada Sedimentary Basin, BSSA". Regionally, they now estimate $\sim$0.8%.**

   *Thank you for bringing this updated rate and reference to our attention. We have updated the reference and value in the text (L 33).*

2. **Line 64: Could the authors also include why only the Z-component was used in the study, and not any horizontals.**

   *We have included a comment here indicating that since a number of the public stations, including R25AM which is used in Fig. 1, are single component sensors, we used the Z-component only in order to be able to compare the noise reduction in a number of different locations (L 72-74).*

3. **Line 80 - Section 3: Please explain why you chose to compare to 2018 and not 2019 in this section. You compared the seismic noise levels to 2019 (lines 77-78), why not also the seismicity?**

   *We have included a statement to clarify this (L 132-133). We are unable to compare seismicity within the KSMMA between 2019 and 2020 as the seismicity from 2019 is not yet published. We are able to analyse the seismic noise since this is only computed at a single station (from which data is available on IRIS), whereas a large number of stations are needed to compure accurate hypocenter locations.*

4. **Line 80 - Section 3: Also, please include somewhere how you processed the time series, any band-pass filter applied? SNR?**

   *We have included this information within Section 3, as requested (L 115-117).*

5. **Line 82-88: More details on the instruments would be good, e.g., sample rates? Are the newly installed stations specifically chosen in some way for certain magnitudes? Site conditions (same for all stations, or different)?**

   *We have extended this paragraph to include more of this information, in particular why the sensors were placed in this array, and the depth of burial of the array (L 96-105). We have also made it more obvious that further information about this array and its installation can be found in Salvage et al., 2021 http://www.geosciencebc.com/summary-of-activities-2020-energy-water/.*

6. **Line 110 - Figure 2: Is the absolute seismicity density the same for the two subplots? I.e. is high density for (a) = high density for (b)? To compare the two periods, it would be good if the colours refer to similar seismicity densities. If not, please make this clear in the caption. Also, is the seismicity used the same as in Figure 3?**

*We have updated Figure 2 to ensure that the absolute seismicity in both a and b are now equal meaning that the high and low density colours can be directly compared between the figures. We have included a comment about this in the figure caption so that the reader is also aware of this. We have updated the caption of Figure 3 to mirror the caption of Figure 2, to make it clearer that this is the same seismicity being plotted. Furthermore, we have updated Figures 2b and 3b to include data to the end of 2020 (which was not available when this paper first went to review).*

7. **Line 113 - Figure 3: Could the authors include the magnitude of completeness for the two different datasets in the figure (to make it easier to compare).**

*We have added this to Figure 3. We have also changed the axes on the plots so that they are the same, allowing a direct comparison of the event counts and magnitudes with time.*

8. **Line 149-150: Could the authors include more details on where the KSM08 station is located? Is it far away from cities/towns? Near any wells?**

*We have included the distance that this station is from the nearest settlement (Rolla), and indicated that the recent seismicity in the vicinity of KSM08 would suggest active wells in the area prior to the lockdown scenario experienced in 2020 (L 174-178).*

9. **Line 153-155 - Figure 4: Any ideas as to why the seismic noise level is low**

**in July at KSM08? Looks to be down at the same levels as during April. Is this a trend seen at more stations than just KSM08?**

*This is seen at a number of stations (to varying degrees). I have confirmed with the regulator that this is a downturn in the market leading to less operations in the area due to company decisions, rather than a government enforced lockdown. We have added a comment about this to the text (L 182-185).*

10.  **Line 165: For the FI value, do you compensate for high-frequency attenuation in some way? You mention that you use one station for all events, won't the low-to-high ratio be different depending on how much high-frequency energy has been attenuated? I.e., events from larger distances have less high-frequency content due to more attenuation than the closer events.**

*We do not compensate for high-frequency attenuation. You are correct in saying that the ratio will be dependent upon attenuation factors, including the distance the event occurs away from the recording station. We stated within the text (L 196-198) that we use station KSM06, which is centrally located in the main clusters of seismic activity. However, we have tested the analysis at all KSM stations and see no temporal patterns within the FI. We have updated the text to make this evident.*

11.  **Line 188-189: Visually, Figure 3b appears to have a slightly decreasing magnitude trend with time. The cloud is around ML 0.0 to 1.0 in June, and ML -0.5 to 0.5 in August. Have you looked into this? Any tests done to find trends?**

*We have carefully looked at the magnitudes with time, in both Figures 3 and 5(a). The potential lowering of the lowest magnitudes from April to August may be an artefact since the last 2 stations in the KSM array were installed in May 2020, thus allowing better azimuthal coverage for event detection and location, as well as lowering the magnitude of completeness. As the magnitudes presented are*

*the average of the magnitude of the event calculated at each station, there is also
a degree of error in the estimation.*

12. **Line 191-204: It would be interesting to see a second plot from before
lockdown and a third after seismicity picked back up again. How does the
b-value change between the three periods?**

*We actually would like to submit this analysis as another paper, showing the effect
of the lockdown and the change in b-values with time in this area, hence we are
not including it here.*

13. **Line 356: Would it be possible for the authors to instead plot each event
as a circle (e.g., based on magnitude as in previous plots) so that they can
highlight the events they identified as triggered by a remote event?**

*The way in which we have determined whether any events have the possibility to
be triggered by remote earthquakes (following the methodology of Wang et al.,
2015), does not allow us to spatially determine which of our detected earthquakes
may have been affected. Instead, it is a statistical measure of the temporal evo-
lution of seismicity before and after the teleseism. For this reason, Fig. 7 is the
best way to present the potential increase in seismic activity within the KSMMA
following a teleseismic event.*

**3   Technical Comments**

- **General technical: figure font size was quite small and needs to be in-
creased.**

*Fonts have been increased on all figures.*

- **Line 7-11: Regarding the three sentences: the authors write that "general
characteristics" are similar between active and shutdown periods, but then**

**go on to state two reasons they are different (magnitude and temporal clusters) and only one reason they are similar (spatially). This makes it seem like they are more different than similar. Perhaps rephrase first sentence.**

*We have re-written this part of the abstract to try and make our meaning more clear and to avoid confusion.*

• **Line 45: Sentence structure is off. "We call this latent seismicity i.e. seismicity..."**

*We have re-written this sentence for clarity.*

• **Line 62: Comma missing: "Following the methodology of Lecocq et al. (2020) we compute the..."**

*This has been added.*

• **Line 72 - Figure 1: Please also explain what the vertical highlighted (yellowish) periods are in the figure caption (occurring before vertical dashed red line). Also, text is very small.**

*The text in all figures has been updated. We have added a comment in the caption about the highlighted periods before the vertical dashed red line, which indicates weekdays.*

• **Line 75-77: Sentence is confusing, consider rearranging: "Following the reopening of some businesses in May and June 2020, the increase in noise is interpreted as the increased movement of people, although it remains lower than pre-pandemic levels." Or something similar.**

*Thank you for this suggestion. We have rephrased this sentence to allow clarity.*

• **Line 87: Why not reference Figure 2 here for the station configuration?**

*We have added a reference to Fig. 2 at this point in the text.*

- **Line 99-101: Sentence doesn't make sense.**

  *We have clarified this sentence.*

- **Line 104-107: In Figure 3 caption, you reference Hutton and Boore (1987) as the origin of your ML calculations. This is not who you reference in the text.**

  *We have changed the structure of the figure caption to reflect that the Hutton and Boore calculation for magnitude was only used for the 2018 catalogue, in work previously carried our by Visser et al. (2020). Our work (seismicity in 2020) uses the magnitude calculation of Babaie-Mahani and Kao (2020), as referenced in the text.*

- **Line 112: "ML 3-4+" doesn't really make sense, either it's ML 3-4, or ML 3+.**

  *We have changed this.*

- **Line 113 - Figure 3: Please use the same y-axis limits on the a) and b) plots since we're supposed to compare them.**

  *The axes of Figure 3 have been updated and now includes all data from 2018 and 2020, which was previously unavailable at the point of submission of this manuscript for review.*

- **Line 114-115: You only state in the Figure 3 caption that the seismicity increase in 2020 March, August, and September are due to hydraulic fracturing operations. Please include this in the text instead of in the figure caption.**

  *This information was already included in the text on lines 113-118, under Section 3 (now L 136-138).*

- **Line 149-150: Comma placement in this sentence is a bit off.**

  *We have re-written this sentence.*

- **Line 156: "a pre-lockdown levels" is grammatically incorrect.**

  *This was supposed to read "as pre-lockdown levels". This has been changed.*

- **Line 161 - Figure 5: Here it says you use the Babaie-Mahani & Kao (2020) formula to compute ML. Same or different to the one in Figure 3?**

  *We have updated the caption for Figure 3 to indicate that it was indeed the same formula as used in Figure 5 (Babaie-Mahani and Kao (2020)).*

- **Line 177-178: "2000 m and 2500 m" please switch to km to stay consistent with previous sentence."**

  *This has been changed.*

- **Figure 6: is not referenced in the text.**

  *This was an oversight on our part. We have added reference to Fig. 6 in the discussion of the Mc and b-value section.*

- **Line 247-249: Strange sentence structure with the commas and parentheses.**

  *This sentence has been re-written.*

- **Line 299: "it" is missing: "in areas affected by hydraulic fracturing it is thought to..."**

  *This has been added.*

---

## Author Response (AR2)

**SE-2020-203: Unprecedented quiescence in resource development area allows detection of long-lived latent seismicity**

**Response to Dr. Stephen Hicks - Topical Editor**

Rebecca O. Salvage (on behalf of the authors)
**Correspondence:** Rebecca O. Salvage (beckysalvage@gmail.com)

*Thank you for these additional comments. In the track changes document, responses to your comments are coloured purple.*

1. **Line 6-9: This is quite a long sentence and has quite a few sub-clauses. Could you maybe consider splitting it up into sentences to make it a bit more readable? Also, I think there is comma missing on Line 8 between "lockdown" and "severely".**

   *We have split these sentences up as you suggest and re-worded them to make the meaning clear.*

2. **Line 43: I don't think "BC" has been defined as an acronym before that point. Also, the full version of "British Columbia" is used in many other places. Please check throughout for consistency.**

   *Thank you for bringing this to our attention. We have changed the instances of BC in the text to the full version for consistency.*

3. **Line 61: Change "resulted due to" to "resulted from".**

   *This has been changed.*

4. **Line 77: Change "including R25AC" to "including station R25AC".**

   *This has been changed.*

5. **Line 100: Please add a couple of commas (highlighted with stars below) to improve readability here. Something like: "Prior to 2020, 9 public sensors maintained by Natural Resources Canada, the BC Oil and Gas Commission, the BC Seismic Research Consortium\*,\* and the Geological Survey of Canada\*,\* existed within the KSMMA boundary, along with 6 co-located accelerometers poised to better capture higher levels of ground motion from larger seismic events".**

   *This has been changed.*

6. **Line 164: Local magnitude values in other places in the paper are given to one decimal place, which reflects the typical error of computed ML. Maybe also give these values to one decimal place? See also Line 209.**

*This has been changed.*

7. **Lines 240-248: Even accounting for Reviewer 2's Specific Comment #1, I find that much of this added text in response to the reviewer is very similar to what is in the first paragraph of Section 3.3. To make the text more concise, maybe you could have another read of both sections to see what is more appropriate for the Results section versus the Discussion section?**

*Thank you for pointing this out to us. We have removed the discussion around the interpretation of the FI in the results section, and only focus on it in the Discussion.*

8. **Line 266: I am not sure what "scenario change" is referring to. Please could you check this.**

*We were referring to the fact that our calculated b-value lies extremely close to 2, which is the cut-off point indicated for a change in the influence of large scale faulting vs. natural fracture systems. We have changed this in the text to simply read "close to 2" to better reflect our meaning here.*

9. **Line 387: Please change "the spatial migration of events" to "the possible spatial migration of events".**

*This has been changed.*

10. **Line 476: Please change "method" to something like either "mechanism" or "process".**

*We have changed this to "scenario".*

11. **Line 500: Please consider changing "the previous trapping of fluids" to "the trapping of fluids previously injected" (if I interpret this correctly!). "Method" to me seems to relate more to a scientific process.**

*We have changed this for clarity.*

12. **Lines 514-516: You say a "low magnitude of completeness" related to lockdown, but in response to Reviewer 1's Comment #5, you also say that you cannot directly compare the pre-lockdown and during-lockdown Mc. However, in Figure 3, you show the Mc = 1.0 for the whole catalogue. So I am wondering if this sentence should be re-jigged a little? It would be neat to have shown a lower Mc during lockdown, but if you were not able to directly compare the two catalogues, then maybe you should add a sentence about this in the paper, rather than just solely responding to the reviewer, as this will be a common question asked by readers of your paper (I suspect)?**

*The Mc shown in Fig. 3(a) reflects the Mc for the year 2018 as calculated by Visser et al. (2020); whereas the Mc shown in Fig. 3(b) shows our calculated Mc of 0.4. We have re-worded this sentence in the conclusions to simply state the Mc, and added a comment in the discussion section about why the Mc in Fig. 3(a) and (b) are not directly comparable.*

13. **Figure 1: Could you please widen this figure to a larger page width so that the text in the legend can be easily read without zooming-in. Please also double-check that the sizes are most suitable for rest of the figures.**

*We followed the LaTeX guidelines for including figures in the manuscript and have made them either 12cm wide (for a full page figure) or 8.3cm wide (for a single column figure), as outlined on Solid Earth's LaTeX template. We have left*

*the figures as they are in this version, as for the final print version individual figure files are provided to Solid Earth,*
*rather than embedded figures in a manuscript.*

55

**SE-2020-203: Unprecedented quiescence in resource development area allows detection of long-lived latent seismicity**

**Response to Reviwer 1 - Dr Calum Chamberlain**

Rebecca O. Salvage (on behalf of the authors)
**Correspondence:** Rebecca O. Salvage (beckysalvage@gmail.com)

*All line numbers refer to line numbers in the updated "clean" manuscript i.e. that without the track changes.*

**1    General Comments**

**This paper documents the detection and analysis of earthquake activity within a normally-active region of hydraulic fracturing during the cessation of activity due to lockdowns associated with the COVID-19 pandemic. This study has an**

5    **interesting and unique position in being able to assess changes in earthquake rate due to a changein hydraulic fracturing activity, alongside changes in earthquake detectability due to the reduction in background seismic noise. The authors find that seismicity during the lockdown does not display the high-rate, temporally clustered sequences otherwise observed associated with reservoir stimulation, however, they do observe consistent seismicity within previously stimulated regions.**

**The authors provide good analysis of possible causes of this non-stimulated seismicity, including discussion of triggering from large regional earthquakes, the impact of pore-pressure, and longer-lasting fluid diffusion and poroelastic effects. The authors finally interpret this "latent" seismicity as being due to an altered stress-state within the previously stimulated regions due to trapped fluids, and infer that these earthquakes are driven by aseismic slip. I would like**

15    **to see more discussion of the interplay between the purported stress-changes, and the strength of the fractures within the reservoir: I wonder if the change in stress is actually the dominant effect, or instead a reduction in fault strength due to prior fracturing would dominate the failure criterion? I'm also curious about how the purported high-pressures are sustained alongside the inter-connectedness of the fracture network?**

20    *These are excellent points, and we have tried to address them in the discussion section of the manuscript (L 400-405; L 451-463). We agree that a change in stress or a reduction in fault strength may be responsible for generating this seismicity, which we did not really discuss before, and so we have added fault weakening into our discussion as a possible cause. We have also added more discussion around how aseismic slip may enable the generation of seismicity beyond the pressurized fluid front (i.e. at large distances from the injection point), as has been hypothesized by a number of authors and have included a*

25  *number of references (e.g. Eyre, 2019; Wei et al., 2015; Cappa et al., 2018). We think that far-field (tectonic) stresses may play a role in sustaining such slip, as well as direct interaction with a pressurized fault patch over shorter distances.*

**Overall I think this is a good, well-written paper documenting an interesting case of reservoir stimulation shutdown. I think it might be relevant to point out the further uniqueness of this study in that most other shutdowns occur either**
30  **after a large event, which would in itself alter the stress state, or when a reservoir is depleted. I have some specific comments below and some minor technical corrections to the manuscript.**

*Thank you for your comments. We have tried to emphasize the more unique aspect of this study throughout the text, and in particular in L 59-64 and L 394-399, as you suggest.*

**2 Specific Comments**

35  1. **Is there any indication of long-lasting changes in the stress-field? e.g. do you observe changes in the stress-ratio or rotation in the principal stress axes associated with the initiation of reservoir stimulation, and is this sustained throughout the background seismicity? I imagine that the seismicity might be too sparse prior to the field becoming active to provide background state, but there may be stress-field data from borehole-breakouts prior to stimulation? I'm also curious about the likely magnitude to stress-variation due to hydraulic fracturing.**

40  *You are quite correct that the seismicity is too sparse prior to the KSMMA becoming active to really detail this. Source characteristics of the events identified as "background" seismicity" are mostly not accounted for. We did find some information regarding borehole breakouts, and a recent study by Babaie-Mahani et al. (2020) which details recent (2018-2019) stresses (from focal mechanism analysis), however a comparison of before and after is not possible due to the limited data. We have added a number of paragraphs into the discussion for this (L 366-393), however at this time*
45  *we are unable to categorically give an answer to this question as it represents a large area of research that is yet to be undertaken.*

2. **Is there any other evidence of aseismic creep? I am not familiar with the paper by Eyre (2020), but I wonder if they found any characteristic temporal evolution of seismicity that they associated with aseismic slip? I'm also**
50  **curious about why an aseismic driver is required? It is not generally assumed that background seismicity requires an aseismic driver, could this not just be the "new-normal" background seismicity after fractures were weakened due to hydraulic fracturing?**

*We have now included more references to aseismic slip in hydraulic fracturing environments, and tried to point out the characteristics that we observe in our seismicity that we believe are indicative of this mechanism (L 473-477). This*
55  *includes the long-lived persistent nature of the seismicity (swarm-like activity rather than a typical mainshock-aftershock sequence), the lack of hypocenter migration (away from a point of injection), as well as dominance of low frequency*

*energy in the waveforms.*

*It is possible that the seismicity detected reflects a new normal in terms of background seismicity, with event counts now greater than they were prior to operations in this area, and we have added this comment in to the manuscript (L 400-405). We were trying to suggest that the generation of this seismicity (whether it is latent and a direct consequence of the recent operations, or whether it is the new background rate) could be generated by aseismic slip (as a result of tectonic forces), or through other mechanisms such as fracture weakening. We have tried to make this more clear in the manuscript by softening our language to advocate that aseismic slip is a proposed mechanism for this seismicity generation, rather than the only mechanism.*

3. **It would be great to have some well-stimulation data to confirm your suspicions in Line 125. I imagine that this is hard to come by, and if so, can you add a note around line 125 to say that well-data were not available.**

*As you note, this information is proprietary between the regulators and the individual companies operating in this area, and as such we cannot comment further on this as we do not have access to this information.*

4. **Around line 267 the authors argue that upwards of 70% of the earthquakes "cannot be explained by this mechanism [natural seismicity]". The argument is based on Gutenberg-Richter scaling, which is quite a weak argument for such a strong statement. Furthermore, the statement of scaling is based on a b-value of 1, which the authors do not find for this region - I'm curious about the reason for using a b-value of 1? Finally: it is hard to make such a strong statement, given that natural earthquake (de-clustered) distributions appear to be almost Poisson and can have variations in rate in time. I suggest more circumspect language in this statement rather than "cannot be explained".**

*We agree that the language here needs to be softened, and we have tried to do so throughout the text in relation to this point. We used a b-value of 1 since this is the expected b-value for natural seismicity in North America (Frohlich and Davis, 1993; Godano et al., 2014), and as we were trying to determine whether any events could be deemed "natural", we chose this. We have added this explanation into the text (L 321-323). It is common to have much higher b-values associated with hydraulic fracturing experiments (as we noted in the text and found in our study), but the aim of this part of the discussion was to determine whether any/some/all of the seismicity may be natural. We have tried to make this clearer.*

5. **I am surprised that your magnitude of completeness appears to have gone up during the shutdown and I'm curious to hear why you think this is? It would also be useful to state what method was used to compute the magnitude of completeness.**

90     *We have added the method we used to calculate the magnitude of completeness (maximum curvature method). We originally included this comment for completeness, however, upon reflection we can see that it is confusing and we have therefore removed the sentence regarding the Mc for the entire catalogue from this paper. We believe that the differing Mc is an artifact due to differing methods used in the magnitude calculations between Salvage et al., 2021 and this paper, meaning they are not directly comparable.*

**95   3   Technical Comments**

Below I have listed my technical comments starting with the relevant line number:

- **23: Change "on such" to "of such"**

  *This was a typo and has been changed.*

- **64: Change "Z-component" to "vertical-component"**

100     *We have changed all references to Z-component to now read the vertical component.*

- **Lines 89-95: While the detection methods are not the key topic of this paper, and it only really matters that the detection method is consistent throughout the study, it would be good to have a little more detail and cite relevant papers - ideally citing another paper using the same methodology would be useful here.**

  *We have added a couple of sentences relating to the methodology employed here, as well as reference to a paper that*
105     *uses it (L 106-114).*

- **134: What was the magnitude of completeness for these precursory events? Saying "A total" suggests absolute completeness.**

  *This was not our intended meaning. We have changed the sentence to reflect that we simply wanted to state the number of events in the precursory sequence. Due to the small number of events in this sequence, the Mc was calculated to be*
110     $\sim 1.1$.

- **175: Remove trailing "a" at the end of the line**

  *This has been removed.*

- **218: "Data is" should be changed to "Data are" - it also isn't clear which data are being referred to here.**

  *We have changed this to be grammatically correct.*

115  - **258: I suggest changing "non-existent" to "undetectable" given the limitations of the seismic networks available.**

  *This has been changed.*

- **314: "Stress field would likely diminish": I suggest rephrasing this, it is hard to imagine the entire stress field diminishing, but there certainly might be a change in orientation and magnitude of principal stress axes. This statement could also do with a citation.**

*We have changed this sentence to be more accurate (and realistic), and included a reference as you suggest (L 446-447).*

- **320: I don't know the paper cited here, but aseismic slip has long been associated with seismicity, so I assume that this paper talks specifically about aseismic slip in hydraulic fracturing environments: It would be good to make that clear in this statement.**

*This has been changed.*

- **365: Change "always" to "since Oil and Gas production started" or similar.**

*This has been changed.*

- **Figure 2(b): The inset and key are not needed as far as I can tell because they repeat from 2(a) - happy to have them left in, but if you do, can you add in the white box on the inset map showing the main figure location I saw this in 2(a), but not 2(b).**

*Figure 2 has been updated to include all seismicity from 2020 (at the time of submission it only included seismicity to October). We have also increased the text size in the figure, and made sure the inset map includes the white box. We have left the legend in both plots.*

**SE-2020-203: Unprecedented quiescence in resource development area allows detection of long-lived latent seismicity**

**Response to Reviwer 2 - Dr Stephen Crane**

Rebecca O. Salvage (on behalf of the authors)
**Correspondence:** Rebecca O. Salvage (beckysalvage@gmail.com)

*All line numbers refer to line numbers in the updated "clean" manuscript i.e. that without the track changes.*

**1   General Comments**

**Overall, this well-written paper establishes the presence of "latent" seismicity in an area where resource development had stopped due to restrictions put in place to prevent the spread of COVID-19. This paper estimates that 70% of the**
5   **earthquakes measured in the Kiskatinaw area, B.C., between April and August 2020, are due to aseismic slip from leftover fluids becoming trapped in the target formation, after the resource development in the area had been paused. The other 30% of seismicity is either background seismicity, which has been occurring before the resource development in the area, or directly triggered from a teleseismic earthquake.**

10   **The authors establish the background of the Kiskatinaw Seismic Monitoring and Mitigation Area (KSMMA), the resource development occurring in the area, and the past and current seismic monitoring taking place. The unique conditions established here involve an expanded seismic network, yet a stoppage of hydraulic fracturing operations due to a global pandemic and not a large seismic event or a depletion of the reservoir as is the case in other areas. The seismic network allows for the detection of small earthquakes, in a region that experiences relatively low natural seismicity.**

**In this paper it is shown that the most of the detected earthquakes in the area between April and August 2020 are: not the result of direct injection, as there are no spatial or temporal clusters around wells and no active wells during this period; not the result of natural seismicity, as the previously measured seismicity rates are too small to account for the number of events detected; and not triggered by large teleseimic earthquakes. This leads the authors to conclude that**
20   **previous fluid injections in the area altered the state of stress in a hydraulically linked formation generating aseismic slip loading on unstable zones.**

**This paper does a good job of testing the possibilities of causes to seismicity in an area after resource development has paused, where seismicity was uncommon beforehand. After testing the seismicity traits against those that are common**

25 from other causes of seismicity, it is determined in this area that the seismicity measured over this period can be labelled as "latent" seismicity. That is, the seismicity is caused by prior resource development in this region, but not directly related to active hydraulic fracturing or salt water disposal.

*Thank you for this nice summary and positive comments.*

**2 Specific Comments**

30 1. **In Section 3.3 the FI index is discussed as way to describe whether fluids play a direct role in seismicity (negative FI) or not (positive FI). I understand that there are no temporal variations of FI, but was a consistently negative FI or consistently positive FI found (Fig 5a)? And would that imply a direct role of fluids or not in the seismicity?**

*We have added a couple of sentences in the discussion where we suggest that the FI is not a commonly used measure in hydraulic fracturing seismicity (it was developed for volcanic environments) and therefore we base the fact that larger*
35 *amounts of low frequency energy within the waveform is indicative of the role of fluid in its generation upon research in the volcanological community (L 223-230).*

2. **Lines 190-193: How is the magnitude of completeness measured? Is it spatially distributed over the KSMMA boundary based on the station density or calculated as the point where the catalogue deviates from the calculated b-value (as shown in Figure 6)? I think there should be a reference to Figure 6 here.**

40 *You are quite correct that we were missing a reference to Fig. 6 at this point. It has now been added. We have also added a comment about how we compute the magnitude of completeness (maximum curvature method) and the $b$-value (least-squares linear regression) for completeness (L 231-237).*

**3 Technical Comments**

- **Line 10 "locate" should be "located"**

45 *This has been changed.*

- **Line 78-79: "In fact, ground displacement remained between 20 and 30 nm at station R25AC for the entirety of 2019." Should this supposed to be the average ground displacement that remained between 20 and 30 nm?**

*This has been changed.*

- **Line 99-101: This sentence needs a proper ending ie. "... previous seismicity (available directly from BCOGC),**
50 **was used to determine the locations in NonLinLoc"**

*Another reviewer also mentioned this, so this sentence and the previous one have been re-written for clarity.*

- **Line 111 and Figure 3 caption: "In both years" appears before the mention of 2018 and 2020. It would be helpful to give the specific years before referring to them as both years.**

*Changes in the figure caption and the text mean that this is no longer the case.*

– **Line 130: "to present" could this now be replaced by a specific date or month?**

*We have now updated the text and the figures to reflect seismic data for the entirety of 2020.*

– **Line 258: CNSN stands for Canadian National Seismograph Network.**

*Thank you for this clarification.*

– **Line 268-270: There are 3 "therefore" in 2 sentences.**

*We have used a number of different synonyms now to avoid our use of "therefore".*

– **Figure 3: I think it would be easier to compare if the two figures had the same x-axis, and the note mentioning that the 2020 data is complete only until October. Comparing the temporal patterns on different time scales seems difficult.**

*The figure has now been updated to represent data for the entirety of 2020, and so both subplots are now on the same x-axis.*

– **Figure 6. Shown but never referenced in the text.**

*This was an oversight on our part and has now been corrected.*

**SE-2020-203: Unprecedented quiescence in resource development area allows detection of long-lived latent seismicity**

**Response to Reviwer 3 - Anonymous**

Rebecca O. Salvage (on behalf of the authors)
**Correspondence:** Rebecca O. Salvage (beckysalvage@gmail.com)

*All line numbers refer to line numbers in the updated "clean" manuscript i.e. that without the track changes.*

**1 General Comments**

**This nicely written and well-structured article examines how seismicity rates during quiescent periods can be used to examine driving factors behind "latent" induced seismicity after direct injection is stopped. The authors study seismicity in the Kiskatinaw area of British Columbia before, during, and after the 2020 COVID-19 government-imposed shutdown to investigate how the seismicity behaves when fluid injection operations in the region temporarily stop. They find evidence for "latent" seismicity during the quiescent period, of which 70% cannot be explained by direct injection, natural seismicity, or triggered earthquakes. After comparing the seismicity trends to those expected from fluid migration models, pore pressure diffusion, poroelastic effects, and aseismic slip, the authors eliminate all but aseismic slip as the most likely driving factor to the latent seismicity. Throughout this process, the authors provide clear analysis and evidence to their arguments.**

**This paper is a unique and timely article with respect to the current pandemic and global lockdowns. The authors apply common and suitable seismological techniques to investigate earthquakes that would otherwise be hidden by direct injection seismicity. I think this article will be of interest to the audience of Solid Earth. Please find the specific and technical comments below.**

*Thank you for these comments - we are glad the reviewer found our article of interest.*

**2 Specific Comments**

1. **Line 28: This number (∼0.3%) has been updated in a later publication: "Ghofrani & Atkinson (2020), Activation Rate of Seismicity for Hydraulic Fracture Wells in the Western Canada Sedimentary Basin, BSSA". Regionally, they now estimate ∼0.8%.**

*Thank you for bringing this updated rate and reference to our attention. We have updated the reference and value in the text (L 33).*

2. **Line 64: Could the authors also include why only the Z-component was used in the study, and not any horizontals.**

*We have included a comment here indicating that since a number of the public stations, including R25AM which is used in Fig. 1, are single component sensors, we used the Z-component only in order to be able to compare the noise reduction in a number of different locations (L 72-74).*

3. **Line 80 - Section 3: Please explain why you chose to compare to 2018 and not 2019 in this section. You compared the seismic noise levels to 2019 (lines 77-78), why not also the seismicity?**

*We have included a statement to clarify this (L 132-133). We are unable to compare seismicity within the KSMMA between 2019 and 2020 as the seismicity from 2019 is not yet published. We are able to analyse the seismic noise since this is only computed at a single station (from which data is available on IRIS), whereas a large number of stations are needed to compute accurate hypocenter locations.*

4. **Line 80 - Section 3: Also, please include somewhere how you processed the time series, any band-pass filter applied? SNR?**

*We have included this information within Section 3, as requested (L 115-117).*

5. **Line 82-88: More details on the instruments would be good, e.g., sample rates? Are the newly installed stations specifically chosen in some way for certain magnitudes? Site conditions (same for all stations, or different)?**

*We have extended this paragraph to include more of this information, in particular why the sensors were placed in this array, and the depth of burial of the array (L 96-105). We have also made it more obvious that further information about this array and its installation can be found in Salvage et al., 2021 http://www.geosciencebc.com/summary-of-activities-2020-energy-water/.*

6. **Line 110 - Figure 2: Is the absolute seismicity density the same for the two subplots? I.e. is high density for (a) = high density for (b)? To compare the two periods, it would be good if the colours refer to similar seismicity densities. If not, please make this clear in the caption. Also, is the seismicity used the same as in Figure 3?**

*We have updated Figure 2 to ensure that the absolute seismicity in both a and b are now equal meaning that the high and low density colours can be directly compared between the figures. We have included a comment about this in the figure caption so that the reader is also aware of this. We have updated the caption of Figure 3 to mirror the caption of Figure 2, to make it clearer that this is the same seismicity being plotted. Furthermore, we have updated Figures 2b and 3b to include data to the end of 2020 (which was not available when this paper first went to review).*

7. **Line 113 - Figure 3: Could the authors include the magnitude of completeness for the two different datasets in the figure (to make it easier to compare).**

*We have added this to Figure 3. We have also changed the axes on the plots so that they are the same, allowing a direct comparison of the event counts and magnitudes with time.*

8. **Line 149-150: Could the authors include more details on where the KSM08 station is located? Is it far away from cities/towns? Near any wells?**

*We have included the distance that this station is from the nearest settlement (Rolla), and indicated that the recent seismicity in the vicinity of KSM08 would suggest active wells in the area prior to the lockdown scenario experienced in 2020 (L 174-178).*

9. **Line 153-155 - Figure 4: Any ideas as to why the seismic noise level is low in July at KSM08? Looks to be down at the same levels as during April. Is this a trend seen at more stations than just KSM08?**

*This is seen at a number of stations (to varying degrees). I have confirmed with the regulator that this is a downturn in the market leading to less operations in the area due to company decisions, rather than a government enforced lockdown. We have added a comment about this to the text (L 182-185).*

10. **Line 165: For the FI value, do you compensate for high-frequency attenuation in some way? You mention that you use one station for all events, won't the low-to-high ratio be different depending on how much high-frequency energy has been attenuated? I.e., events from larger distances have less high-frequency content due to more attenuation than the closer events.**

*We do not compensate for high-frequency attenuation. You are correct in saying that the ratio will be dependent upon attenuation factors, including the distance the event occurs away from the recording station. We stated within the text (L 196-198) that we use station KSM06, which is centrally located in the main clusters of seismic activity. However, we have tested the analysis at all KSM stations and see no temporal patterns within the FI. We have updated the text to make this evident.*

11. **Line 188-189: Visually, Figure 3b appears to have a slightly decreasing magnitude trend with time. The cloud is around ML 0.0 to 1.0 in June, and ML -0.5 to 0.5 in August. Have you looked into this? Any tests done to find trends?**

*We have carefully looked at the magnitudes with time, in both Figures 3 and 5(a). The potential lowering of the lowest magnitudes from April to August may be an artefact since the last 2 stations in the KSM array were installed in May 2020, thus allowing better azimuthal coverage for event detection and location, as well as lowering the magnitude of completeness. As the magnitudes presented are the average of the magnitude of the event calculated at each station, there is also a degree of error in the estimation.*

12. **Line 191-204: It would be interesting to see a second plot from before lockdown and a third after seismicity picked back up again. How does the b-value change between the three periods?**

*We actually would like to submit this analysis as another paper, showing the effect of the lockdown and the change in b-values with time in this area, hence we are not including it here.*

13. **Line 356: Would it be possible for the authors to instead plot each event as a circle (e.g., based on magnitude as in previous plots) so that they can highlight the events they identified as triggered by a remote event?**

*The way in which we have determined whether any events have the possibility to be triggered by remote earthquakes (following the methodology of Wang et al., 2015), does not allow us to spatially determine which of our detected earthquakes may have been affected. Instead, it is a statistical measure of the temporal evolution of seismicity before and after the teleseism. For this reason, Fig. 7 is the best way to present the potential increase in seismic activity within the KSMMA following a teleseismic event.*

**3 Technical Comments**

– **General technical: figure font size was quite small and needs to be increased.**

*Fonts have been increased on all figures.*

– **Line 7-11: Regarding the three sentences: the authors write that "general characteristics" are similar between active and shutdown periods, but then go on to state two reasons they are different (magnitude and temporal clusters) and only one reason they are similar (spatially). This makes it seem like they are more different than similar. Perhaps rephrase first sentence.**

*We have re-written this part of the abstract to try and make our meaning more clear and to avoid confusion.*

– **Line 45: Sentence structure is off. "We call this latent seismicity i.e. seismicity..."**

*We have re-written this sentence for clarity.*

– **Line 62: Comma missing: "Following the methodology of Lecocq et al. (2020) we compute the..."**

*This has been added.*

– **Line 72 - Figure 1: Please also explain what the vertical highlighted (yellowish) periods are in the figure caption (occurring before vertical dashed red line). Also, text is very small.**

*The text in all figures has been updated. We have added a comment in the caption about the highlighted periods before the vertical dashed red line, which indicates weekdays.*

– **Line 75-77: Sentence is confusing, consider rearranging: "Following the reopening of some businesses in May and June 2020, the increase in noise is interpreted as the increased movement of people, although it remains lower than pre-pandemic levels." Or something similar.**

*Thank you for this suggestion. We have rephrased this sentence to allow clarity.*

– **Line 87: Why not reference Figure 2 here for the station configuration?**

*We have added a reference to Fig. 2 at this point in the text.*

– **Line 99-101: Sentence doesn't make sense.**

*We have clarified this sentence.*

– **Line 104-107: In Figure 3 caption, you reference Hutton and Boore (1987) as the origin of your ML calculations. This is not who you reference in the text.**

*We have changed the structure of the figure caption to reflect that the Hutton and Boore calculation for magnitude was only used for the 2018 catalogue, in work previously carried our by Visser et al. (2020). Our work (seismicity in 2020) uses the magnitude calculation of Babaie-Mahani and Kao (2020), as referenced in the text.*

– **Line 112: "ML 3-4+" doesn't really make sense, either it's ML 3-4, or ML 3+.**

*We have changed this.*

– **Line 113 - Figure 3: Please use the same y-axis limits on the a) and b) plots since we're supposed to compare them.**

*The axes of Figure 3 have been updated and now includes all data from 2018 and 2020, which was previously unavailable at the point of submission of this manuscript for review.*

– **Line 114-115: You only state in the Figure 3 caption that the seismicity increase in 2020 March, August, and September are due to hydraulic fracturing operations. Please include this in the text instead of in the figure caption.**

*This information was already included in the text on lines 113-118, under Section 3 (now L 136-138).*

– **Line 149-150: Comma placement in this sentence is a bit off.**

*We have re-written this sentence.*

– **Line 156: "a pre-lockdown levels" is grammatically incorrect.**

*This was supposed to read "as pre-lockdown levels". This has been changed.*

– **Line 161 - Figure 5: Here it says you use the Babaie-Mahani & Kao (2020) formula to compute ML. Same or different to the one in Figure 3?**

*We have updated the caption for Figure 3 to indicate that it was indeed the same formula as used in Figure 5 (Babaie-Mahani and Kao (2020)).*

– **Line 177-178: "2000 m and 2500 m" please switch to km to stay consistent with previous sentence."**

*This has been changed.*

– **Figure 6: is not referenced in the text.**

*This was an oversight on our part. We have added reference to Fig. 6 in the discussion of the Mc and b-value section.*

– **Line 247-249: Strange sentence structure with the commas and parentheses.**

*This sentence has been re-written.*

– **Line 299: "it" is missing: "in areas affected by hydraulic fracturing it is thought to..."**

*This has been added.*